An integrative taxonomic analysis reveals a new species of lotic Hynobius salamander from Japan

Okamiya Hisanori 1
Sugawara Hirotaka 1
Nagano Masahiro 2
http://orcid.org/0000-0002-7576-2283 Poyarkov Nikolay A. 3 n.poyarkov@gmail.com
1 Department of Biology, Faculty of Science, Tokyo Metropolitan University , Tokyo , Japan
2 Faculty of Science and Technology, Oita University , Dannoharu, Oita , Japan
3 Department of Vertebrate Zoology, Faculty of Biology, Lomonosov Moscow State University , Moscow , Russia
Wallis Graham
Electronic publication date: 2018 Jun 21
Publication date: 2018
Volume: 6
Electronic Location ID: e5084
Received 2018 May 2; Accepted 2018 Jun 5
Copyright: © 2018 Okamiya et al.
Copyright year: 2018
Copyright holder: Okamiya et al.
License: This is an open access article distributed under the terms of the Creative Commons Attribution License, which permits unrestricted use, distribution, reproduction and adaptation in any medium and for any purpose provided that it is properly attributed. For attribution, the original author(s), title, publication source (PeerJ) and either DOI or URL of the article must be cited.
License URL: https://creativecommons.org/licenses/by/4.0/

Keywords: Caudata, Hynobiidae, Honshu, mtDNA, Morphometry, Biogeography, Hynobius kimurae, Hynobius boulengeri, Hynobius fossigenussp. nov., RAG1

Funding: Russian Science Foundation RSF grant No. 14-50-00029 to Nikolay A. Poyarkov Specimen storage and examination was completed with the financial support of the Russian Science Foundation (RSF grant No. 14-50-00029) to Nikolay A. Poyarkov. The funders had no role in study design, data collection and analysis, decision to publish, or preparation of the manuscript.

==============================
We examine the phylogenetic structure and morphological differentiation within the Hynobius kimurae–H. boulengeri species complex—a widely-distributed group of stream-breeding hynobiid salamanders, inhabiting montane areas of western, central and eastern parts of Honshu Island, Japan. Phylogenetic relationships were assessed based on analyses of mitochondrial (mtDNA) and nuclear (nuDNA) gene fragments for a total of 51 samples representing 23 localities covering the entire range of the species complex. Morphological study included one-way analysis of variance (ANOVA) and principal components analysis (PCA) analyses of 26 morphometric and six meristic characters for 38 adult specimens of H. kimurae and three adult specimens of H. boulengeri. MtDNA genealogy supported monophyly of the H. kimurae–H. boulengeri complex, which is sister to all other Hynobius except H. retardatus. The complex is subdivided into three major clades, corresponding to the Eastern (Clade I) and Western (Clade II) populations of H. kimurae sensu lato, and to H. boulengeri (Clade III). Monophyly of H. kimurae sensu lato is only moderately supported by mtDNA, while nuDNA suggested that the Western form of H. kimurae is closer to H. boulengeri than to the eastern form. The time of the split of the H. kimurae–H. boulengeri complex is estimated as late Miocene and coincides with intensive crust movement in western Japan. Divergence between Clades I and II took place in early Pliocene and was likely influenced by the uplift of Central Japanese Highlands. All three clades were found to be different in a number of morphological characters, allowing us to describe the eastern form of H. kimurae as a new species, Hynobius fossigenus sp. nov.

Introduction

Stream-dwelling freshwater organisms are promising models for studies of the interdependence between geological history and formation of biota. Due to their limited dispersal abilities salamanders are readily isolated by geographical barriers such as mountains, river valleys and sea straits (Avise, 2000; Tominaga et al., 2006; Poyarkov et al., 2012). A number of seminal works on freshwater fish, for example, demonstrated significant structuring within wide-ranged species complexes distributed across Honshu Island of Japan that suggest the presence of cryptic lineages in the northeastern and southwestern parts of the island (Machida et al., 2006; Watanabe et al., 2006; Tominaga, Nakajima & Watanabe, 2016). These lineages are often separated by Fossa Magna: the tectonic fault zone crossing the center of what currently is the Honshu Island, which divided the southwestern and northeastern paleo-Japanese landmasses in the past (Kato, 1992).

The genus Hynobius Tschudi, 1838 is the largest genus in the Asian salamander family Hynobiidae currently including 38 recognized species (AmphibiaWeb, 2018; Frost, 2018; Sugawara et al., 2018). Salamanders of the genus Hynobius are traditionally assigned to two major eco-morphological groups: lentic species that breed in still waters and lay large number of small pigmented eggs per clutch, and lotic species that reproduce in mountain streams and lay few larger unpigmented eggs (Sato, 1943). While lentic Hynobius are widely distributed across the Japanese Islands, as well as in the mainland East Asia (Korea and China), lotic species are confined to montane regions of Japan and Taiwan. Recent studies led to the discovery of a number of new species within lotic Hynobius both in Japan and in Taiwan, indicating that taxonomic diversity of these salamanders might still be underestimated (Matsui et al., 2004; Lai & Lue, 2008; Tominaga & Matsui, 2008; Nishikawa & Matsui, 2014; Matsui, Nishikawa & Tominaga, 2017).

Hynobius kimurae Dunn, 1923, also known as Hida salamander, is the most widespread among all lotic Hynobius species, occurring in highland areas of the eastern, central and western parts of Honshu, the main island of Japan, including Kanto, Chubu, Kinki and Chugoku Districts (Matsui, 1981) (see Fig. 1). For a brief review on the history of taxonomic studies of H. kimurae, see File S1. The first study on variation within H. kimurae was carried out by Sato (1933), who examined morphological variation within a population of Tango Peninsula, central Honshu, and demonstrated significant sexual dimorphism and intrapopulational variation in the shape of vomerine tooth series (VTS), presence and size of the fifth toe and in coloration pattern. Later in his milestone monograph, Sato (1943) recognized seven types of dorsal coloration in H. kimurae (ranging from completely dark dorsum with no yellow spots to numerous confluent spots that form a continuous light dorsal pattern) and four types of VTS (from V-shaped to shallow U-shaped). However, he did not analyze the geographic variation of these characters, and most of his specimens were collected from central and western parts of Honshu.

Figure 1 Map of central and western parts of Honshu Island, Japan, showing sampling localities of Hynobius kimurae sensu lato (closed circles: 1–23) and H. boulengeri (closed circle: 24).

Blue and red areas indicate known distribution range of H. kimurae sensu lato (Western and Eastern groups, respectively), the black area corresponds to the known distribution range of H. boulengeri. A dot in a center of a circle denotes the type locality. Study sites: ISTL, Itoigawa-Shizuoka Tectonic Line; JMTL, Japanese Median Tectonic Line. For locality information see Tables S1 and S2.

Ikebe, Yamamoto & Kohno (1986) were the first to analyze chromosome variation in H. kimurae, and they reported the presence of two distinct karyotypes in this species. They showed that H. kimurae and H. boulengeri have similar karyotypes with 2n = 58, which indicates close phylogenetic relationships between these two species. Further study on chromosome variation in H. kimurae (Ikebe & Kohno, 1991) showed the presence of a distinct karyotype in the eastern populations from Kanto District (Tokyo, type 1), which differed from the populations of central Honshu, Chubu District (Ishikawa Prefecture, type 2) in morphological features of the chromosome Nos. 10 and 15–29 and in C-banding patterns, suggesting significant differentiation between the two. Consequent thorough studies on the natural history of the eastern (Kanto District) and western (Kinki District) populations of H. kimurae indicated that they significantly differ in egg number per clutch and larval biology (Misawa & Matsui, 1997), as well as in growth pattern (Misawa & Matsui, 1999).

A detailed study of biochemical variation in H. kimurae was undertaken by Matsui et al. (2000), who performed allozyme analysis of 21 populations across the species range. They showed that the H. kimurae range is subdivided into three major highly genetically divergent groups of populations: (1) Eastern, (2) Central and (3) Western group, with the two latter being more closely related to each other than to the Eastern group. Consequently, a study by Matsui, Misawa & Nishikawa (2009) examined 24 morphological characters from 24 populations across the H. kimurae range and demonstrated the separation of the species into two major geographic groups of populations: (1) Eastern group (localities from the western part of Kanto District to Shizuoka Prefecture in the eastern part of Chubu District), and (2) Western group (localities from Aichi Prefecture in the western part of Chubu District westwards to Kinki and Chugoku districts). These results clearly suggest that taxonomy of the H. kimurae species complex requires reconsideration. However, most of the abovementioned studies compared H. kimurae to the distantly related H. naevius, but did not include H. boulengeri in the analyses. Moreover, no analysis of phylogenetic structure throughout the wide distribution range of H. kimurae based on DNA-markers has been conducted so far.

In the present paper, we use an integrative taxonomic approach, analyzing the variation of mitochondrial (mtDNA) and nuclear (nuDNA) genetic markers, adult and egg sac morphology, as well as available evidence from previous studies, to assess phylogenetic relationships and taxonomy of the H. kimurae–H. boulengeri species complex, and discuss the historical biogeography of this group of salamanders.

Materials and Methods

Species concept

In the present study, we follow the general lineage concept (GLC: De Queiroz, 2007) which proposes that a species is a population of organisms that evolves independently from other such populations due to a lack of gene flow (Barraclough, Birky & Burt, 2003; De Queiroz, 2007). However, integrative studies on the nature and origins of species are increasingly using a wider range of empirical data to delimit species boundaries, rather than relying solely on traditional taxonomic procedure (Coyne & Orr, 1998; Knowles & Carstens, 2007; Fontaneto et al., 2007; Feulner et al., 2007; Leaché et al., 2009). Under the GLC herein, we follow the framework of integrative taxonomy (Padial et al., 2010; Vences et al., 2013) that combines multiple independent lines of evidence to assess the taxonomic status of the lineages in question: mtDNA-based molecular genealogies were used to infer species boundaries, univariate (ANOVA) and multivariate (PCA) morphological analyses were used to describe those boundaries, while nuDNA-marker was analyzed to test the concordance between phylogenetic signals from mtDNA-based genealogy and nuDNA-based phylogeny.

Nomenclatural acts

The electronic version of this article in portable document format will represent a published work according to the International Commission on Zoological Nomenclature (ICZN), and hence the new names contained in the electronic version are effectively published under that Code from the electronic edition alone (see Articles 8.5–8.6 of the Code). This published work and the nomenclatural acts it contains have been registered in ZooBank, the online registration system for the ICZN. The ZooBank Life Science Identifiers (LSIDs) can be resolved and the associated information can be viewed through any standard web browser by appending the LSID to the prefix http://zoobank.org/. The LSID for this publication is as follows: urn:lsid:zoobank.org:pub:AE462D10-3947-445D-8B3B-090675EDBA91. The online version of this work is archived and available from the following digital repositories: PeerJ, PubMed Central and CLOCKSS.

Sampling strategy

Field work was conducted from January 28, 2007 to March 1, 2018 on Honshu Island, Japan. Geographic coordinates and elevation were obtained using Garmin GPSMAP 60CSx and recorded in WGS 84 datum. For the genetic study we collected a total of 51 samples from larval and adult specimens of H. kimurae sensu lato from 23 localities covering the entire distribution of the species, including the type locality (Mt. Hieizan, Kyoto) (populations 1–23; see Fig. 1; Table S1). Our genetic sampling covers eight populations from Kanto District (localities 1–8), nine populations from Chubu District (localities 9–17), four populations from Kinki District (localities 18–21) and two populations from Chugoku District (localities 22–23). For genetic comparisons, one individual of H. boulengeri was collected at Wakayama Prefecture (population 24; Fig. 1; Table S1). Details on geographic locations of studied populations are presented in Table S2. For the morphological study, a total of 38 adult specimens were collected from the eastern (Hinode-san Mt., Hinode; 26 specimens) and western parts (Kumogahata, Kyoto; 12 specimens) of the H. kimurae range. After collection, specimens were photographed in life before being fully anesthetized by tricaine methanesulfonate (MS-222) solution and euthanized. Tissue samples for genetic analysis were taken prior to preservation and stored in 95% ethanol. Specimens were fixed in 4% formalin for 24 h with subsequent storage in 70% ethanol. Specimens and tissues were subsequently deposited in the herpetological collections of the Yokosuka City Museum (YCM, Yokosuka, Kanagawa Province, Japan) and the Zoological Museum of Moscow University (ZMMU, Moscow, Russia). For comparison purposes, we examined lotic-breeding Hynobius specimens stored in the collections of ZMMU and in the National Science Museum, Tokyo (NSMT) (see File S2 for details).

Specimen collection protocols and animal operations followed the Institutional Ethical Committee of Animal Experimentation of the Tokyo Metropolitan University (certificate number 20 of April 1, 2007) and strictly complied with the ethical conditions of the “Guidelines for proper implementation of animal experiments” by the Science Council of Japan (June 1, 2006). Field work, including collection of samples and animals in the field, was authorized by the education board of Utsunomiya, Tochigi, Japan (permit number 102 of April 22, 2014 issued to F. Hayashi and H. Sugawara).

DNA isolation, PCR conditions, and sequencing

Firstly, we obtained partial sequence data of 16S rRNA and cytochrome b (cyt b) mtDNA genes from tissue samples preserved in 95% ethanol. Total genomic DNA was extracted from samples using a DNeasy Blood and Tissue Kit (Qiagen, Hilden, Germany). DNA amplification was performed in 20 μl reactions using ca. 50 ng genomic DNA, 10 nmol of each primer, 15 nmol of each dNTP, 50 nmol of additional MgCl2, Taq PCR buffer (10 mM of Tris–HCl, pH 8.3, 50 mM of KCl, 1.1 mM of MgCl2, and 0.01% gelatine) and 1 U of Taq DNA polymerase. We used primers 16L-1 and 16H-1 to amplify 16S rRNA fragments (Hedges, 1994), and primers HYD-Cytb-F1 and HYD-Cytb-R1 to amplify cyt b fragments (Matsui et al., 2008) for the 52 samples of the H. kimurae–H. boulengeri species complex (Table S1; for primer sequences see File S3). PCR protocols for 16S rRNA and cyt b gene fragments followed Hedges (1994) and Matsui et al. (2008), respectively. Secondly, we obtained partial sequence data for the nuDNA gene RAG1 (recombination activating protein 1) for 32 selected samples that represent main genetic groups in the mtDNA genealogical tree and one sample of H. boulengeri (see Table S1 for details). RAG1_F_N1 and RAG1_R_N3 primers used in PCR followed Nishikawa et al. (2013) (see File S3 for details). The PCR protocol was as follows: an initial 3 min denaturing step at 94 °C, 39 thirty second cycles at 94 °C, 45 s at 56 °C, and 90 s at 72 °C, with a final 10 min extension at 72 °C. The PCR products were purified with Illustra™ ExoStar™ 1-Step (GE Healthcare, Buckinghamshire, UK) and sequenced using BigDye® Terminator ver. 3.1 (Applied Biosystems, Foster City, CA, USA) on an ABI 3130xl Genetic Analyzer (Applied Biosystems, Foster City, CA, USA). Primers16S 16L-1, HYD-Cytb-F1, and RAG1_F_N1 were used as the cycle sequencing primers for 16S rRNA, cyt b, and RAG1, respectively.

Phylogenetic analyses

Sequences were aligned using the Clustal W algorithm (Thompson, Higgins & Gibson, 1994) in BioEdit Sequence Alignment Editor 7.1.3.0 (Hall, 1999), with default parameters. Subsequently, the alignment was checked and manually revised if necessary using Seqman 5.06 (Burland, 1999). Genetic distances and sequence statistics were calculated using MEGA7 (Kumar, Stecher & Tamura, 2016). We deposited the resultant sequences in GenBank (accession numbers MH253618–MH253668 and MH287353–MH287433; see Table S1).

We added GenBank data for H. kimurae (AB266674 and AB201705) and H. boulengeri (AB201706 and AB26675) from environs of species’ type localities, as well as 18 species of Hynobius salamanders (see Table S1), to our analyses. H. retardatus (AB363609) was used as an outgroup species, since this species was shown to be a sister taxon to all remaining members of the genus Hynobius (Zheng et al., 2011, 2012; Weisrock et al., 2013; Chen et al., 2015). In order to assess evolution of lotic and lentic life histories in Hynobius, we ran analyses of an enlarged sampling which covered all major species groups of the genus (see Table S3). We concatenated the sequence data of 16S rRNA and cyt b. The best-fit substitution models were determined with PartitionFinder ver. 2.1.1 (Lanfear et al., 2017), using the Akaike Information Criterion (Akaike, 1974) and the Bayesian Information Criterion (Schwarz, 1978) for Maximum Likelihood (ML) and Bayesian Inference (BI) analyses, respectively. 16S rRNA gene was treated as a single partition, cyt b was partitioned by codon position.

ML trees were generated using TREEFINDER ver. 2011 (Jobb, Haeseler & Strimmer, 2004). Nodal support was estimated with 1,000 bootstrap replicates (ML BS). We a priori regarded tree topologies with ML BS of 75% or greater as sufficiently supported (Huelsenbeck & Hillis, 1993). BI trees were generated using MrBayes 3.1.2 (Ronquist & Huelsenbeck, 2003). BI analyses were performed using three heated and one cold Metropolis coupled Markov Chain Monte Carlo for 200 million generations, with sampling every 100 generations. We checked the convergence of the runs and that the effective sample sizes were all above 200 by exploring the likelihood plots using TRACER ver. 1.6 (Rambaut & Drummond, 2013). The initial 10% of trees were discarded as burn-in. After burn-in, trees of two independent runs were combined in a single majority consensus topology. Confidence in BI tree topology was assessed using posterior probability (BI PP) (Huelsenbeck & Ronquist, 2001). We a priori considered BI PP 0.95 or greater as significant support (Leaché & Reeder, 2002). The allele network for the RAG1 gene was constructed using median-joining method in the PopArt ver. 1.5 (Leigh & Bryant, 2015) with 95% connection limit.

Divergence time estimates

Molecular divergence dating was performed in BEAST 1.8.0 (Drummond et al., 2012), using the mtDNA concatenated dataset. There is no known reliable fossil record for Hynobius. We used the following recently estimated calibration lognormal priors as calibration points to estimate ages (data from Chen et al., 2015): (a) basal split of Hynobius (ca. 13.9–19.7 millions of years (MYA)); (b) split between the common ancestor of the H. kimurae–H. boulengeri species complex and the remaining Hynobius (ca. 13.5–19.2 MYA); and (c) split between the common ancestor of H. lichenatus–H. tokyoensis group and the remaining Hynobius (ca. 8.6–12.6 MYA). Divergence times were estimated using the optimal partitioning strategy determined by PartitionFinder and normal prior distributions. BEAST analyses were performed using the lognormal relaxed clock algorithm (Drummond et al., 2006), and the best topology obtained from the BI analyses was used as a starting tree. We used default prior distributions for all other parameters and ran the analyses for 200 million generations, with samples taken every 1,000 generations under an uncorrelated lognormal relaxed clock and a Yule tree prior. Suitable burn-in and convergence of the parameters were assessed using TRACER ver. 1.6 with Effective Sample Size (ESS) values >200 taken as evidence for convergence.

Morphological characteristics and analyses

In total, 38 adult specimens of H. kimurae (12 specimens from Kumogahata, Kyoto, 26 specimens from Hinode Mt., Tokyo) and three adult specimens of H. boulengeri (from Wakayama and Nara Prefectures) were subjected to morphological examination. Specimens were examined for 26 morphometric and six meristic characters that are widely used in salamander taxonomy and species determination in Hynobius following Nishikawa et al. (2007) and Matsui, Misawa & Nishikawa (2009) (see File S4, A and B for details). For holotype description we additionally examined 10 morphometric characters (following Nishikawa et al., 2007; Poyarkov et al., 2012) (File S4, C). For larval specimens, we recorded nine morphometric characters (see File S4, A for details). Description of the egg sac morphology was performed following Nishikawa, Sato & Matsui (2008) (see File S4, D for details). All measurements were taken rounded to the nearest 0.1 mm using digital calipers; small absolute values (<10 mm) were measured under a stereoscopic binocular microscope (SZ-ST, Olympus, CO), which was also used to count the number of teeth. Measurements and counts of bilateral morphological structures were taken for each side (in right, then left order). Costal grooves were counted excluding axillary and inguinal grooves following Misawa (1989). The number of individuals with regenerated tails was also recorded. Developmental stages were determined following the Akita (2001) tables of normal development for H. kimurae.

For a Tokyo-C population of H. kimurae sensu lato (locality 6 in Fig. 1), which contained both sexes in sufficient numbers for comparison, we examined sexual difference in morphometric characters via the analysis of covariance using Snout-Vent Length (SVL) as an independent variable with Tukey-like tests (Zar, 1984). We used Student’s t-test to assess sexual differentiation in SVL; sexual differences in meristic characters and character ratios (R) were assessed using a Mann–Whitney U-test.

Among the examined populations, SVL was compared using one-way ANOVA with Tukey–Kramer test. The percentage ratio (R) of each morphometric character to SVL was subsequently calculated; we compared 25 character ratios against SVL and six meristic characters among populations using the Kruskal–Wallis test. Tail length (TAL) and fifth toe length (5TL) were omitted from the analyses since some individuals completely lacked fifth toe or had regenerated tails. PCA was conducted to examine overall morphological variation among populations using loge-transformed metric values. When a high correlation between certain pairs of characters was found, we omitted one of them from the analyses to exclude possible overweighting effects. Additionally, we compared certain character ratios, which are considered to be taxonomically important in Hynobius, such as VTS width to length (VTW/VTL), tail width to height in its middle region (MTAW/MTAH), and tail width to maximal tail height (MTAW/MXTAH). Statistical analyses were carried out using Statistica 8.0 (Version 8.0; StatSoft, Tulsa, OK, USA). The significance level was P < 0.05.

Results

Sequence variation

For mtDNA, we obtained 458–549 bp fragment of 16S rRNA, and 628–1138 bp fragment covering partial sequences of tRNAGlu and cyt b genes. We obtained sequences only from cyt b gene for one sample and only from 16S rRNA gene for five samples (see Table S1). The final concatenated alignment of mtDNA data contained 2,345 bp, of which 232 sites were variable and 117 sites were parsimony informative within the in group. For nuDNA, we obtained 449–518 bp fragment of RAG1gene, of which 12 sites were variable and eight sites were parsimony informative. The number of heterozygous sites was zero.

Phylogenetic inference from mtDNA data

For ML and BI analyses, the best partitioning scheme and models of nucleotide substitutions were: 16S rRNA and tRNAGlu (GTR + I + G), 1st position of cyt b (TrN), 2nd position of cyt b (HKY + I), 3rd position of cyt b (HKY).

Phylogenetic analyses employing two different approaches (BI and ML) yielded nearly identical topologies that slightly differed only in associations at several poorly supported nodes (Fig. 2; Fig. S1). Phylogenetic relationships among Hynobius lineages were sufficiently resolved with several strongly supported major nodes (1.0/100, hereafter node support values are given for BI PP/ML BS, respectively; Fig. 2). Our analyses support the monophyly of the group that includes H. boulengeri and H. kimurae sensu lato (1.0/100) and its sister position to all other Hynobius species except H. retardatus (Fig. 2). According to our data, H. kimurae is distantly related to H. naevius, while H. boulengeri has no phylogenetic affinity with species which were previously confused with or considered to be a part of H. boulengeri sensu lato (including H. hirosei Lantz, 1931, H. shinichisatoi Nishikawa & Matsui, 2014, H. osumiensis Nishikawa & Matsui, 2014, H. amakusaensis Nishikawa & Matsui, 2014, H. ikioi Matsui, Nishikawa & Tominaga, 2017 and H. katoi Matsui, Kokuryo, Misawa, & Nishikawa, 2004; see Fig. S1). MtDNA genealogy suggests that the H. kimurae–H. boulengeri species complex is subdivided into three major highly-divergent clades (Fig. 2): (1) Clade I includes the populations of the eastern part of the H. kimurae sensu lato range (from Kanto District to the eastern part of Aichi Prefecture in Chubu District, see Fig. 1); (2) Clade II includes the populations from the central and western part of the H. kimurae sensu lato range (from the northern part of Aichi Prefecture in Chubu District westwards to Chugoku District, see Fig. 1); (3) Clade III includes two populations of H. boulengeri from Kii Peninsula in Kinki District. Monophyly of H. kimurae sensu lato (Clades I + II) was only moderately supported in our analyses (0.94/87; Fig. 2).

Figure 2 BI genealogy of Hynobius kimurae sensu lato and related species reconstructed from 16S rRNA and cyt b sequences.

Values on the branches correspond to BI PP/ML BS, respectively; black and white circles correspond to well-supported (BI PP ≥ 0.95; ML BS ≥ 90) and moderately supported (0.95 > BI PP ≥ 0.90; 90 > ML BS ≥ 75) nodes, respectively; no circles indicate unsupported nodes. Color marking of species in H. kimurae–H. boulengeri species complex corresponds to Figs. 1 and 3–5. For locality information see Tables S1 and S2. Photos by H. Okamiya and N.A. Poyarkov.

Within H. kimurae sensu lato, significant geographic structuring was observed both in Clades I and II. The Eastern group (Clade I) was subdivided into two subclades: I-1 included the populations from Kanto District (populations 1–8 in Fig. 1), while I-2 included the populations from the eastern Chubu District (populations 9–11 in Fig. 1). Phylogenetic position of the population from the eastern part of Aichi Prefecture (locality 12 in Fig. 1) was unresolved. However, in the BI tree used for BEAST analysis it was placed within subclade I-2 (see below). Within the Western Clade II, three main subclades were suggested by mtDNA genealogy: II-1 included the populations from the central part of the species range in Chubu and Kinki districts (populations 13–15, 18–19 in Fig. 1, see also Fig. S1) and also included the topotype population of H. kimurae (Mt. Hieizan, locality 18); II-2 included the populations from the western part of the species range in Chugoku and western Kinki districts (populations 20–23 in Fig. 1); and II-3 included two populations from the northern limit of the H. kimurae range in the northern part of Chubu District (populations 16–17 in Fig. 1). Genealogical relationships within Clade II are strongly supported, suggesting that the subclade II-3 is a sister lineage to subclades II-1 + II-2 (1.0/91; Fig. 1).

Genetic divergence in mtDNA-markers

The uncorrected genetic P-distances for 16S rRNA and cyt b gene fragments among and within the studied Hynobius species are presented in the Table S4. Genetic divergence between the three clades of the H. kimurae–H. boulengeri species complex varied from 1.69% to 2.96% for16S rRNA gene (between H. kimurae Clade II-1–H. boulengeri Clade III, and H. kimurae Clade I-1–H. boulengeri Clade III, respectively), and from 5.98% to 6.65% for cyt b gene (between H. kimurae Clade I-2–H. kimurae Clade II-2, and H. kimurae Clade II-2–H. boulengeri Clade III, respectively). Genetic divergence within clades was notably lower (below 1.15% for 16S rRNA and below 6.20% for cyt b gene) (see Table S4).

Genetic differentiation according to nuDNA data

The analysis yielded nine haplotypes of RAG1 in the H. kimurae–H. boulengeri species complex. Despite the generally low divergence of the examined partial RAG1-fragment (overall mean P-distance 0.65%), the resulted allele network shows clear geographic structuring. The haplotypes were separated into two major groups: the Eastern group had four unique haplotypes of RAG1 gene, which diverged from all other haplotypes observed in the Western group of H. kimurae and H. boulengeri by four mutation steps (Fig. 3). Furthermore, RAG1 allele of H. boulengeri was found to be very closely related to the main haplotype of the Western group of H. kimurae, being separated from it by only a single mutation step (Fig. 3). The haplotype of H. kimurae from Ishikawa (population 17 in Fig. 1) was more divergent, separated by three mutation steps from the main haplotype of the Western group. These results support evolutionary distinction among the Eastern and Western groups of H. kimurae, and suggest a mito-nuclear discordance in phylogenetic placement of H. boulengeri.

Figure 3 Nuclear allele median-joining network for the RAG-1 gene haplotypes observed in Hynobius kimurae sensu lato and H. boulengeri.

Circle sizes are proportional to the number of samples/sequences, small open circles indicate hypothetical haplotypes (alleles). The color of the circles in the network corresponds to the color of mitochondrial lineages in Figs. 1 and 2.

Divergence times estimates

Estimated node-ages and the 95% highest posterior density (95% HPD) for the main nodes are detailed in Table 1. Our analysis suggests that the ancestors of the H. kimurae–H. boulengeri species complex diverged in the end of Miocene ca. 7.0 MYA (5.0–9.3), and the basal radiation of H. kimurae sensu lato was estimated to have occurred around the Miocene–Pliocene border 5.2 MYA (3.7–6.9) (Fig. 4). Basal radiation within each clade of the H. kimurae–H. boulengeri species complex was dated as mid-Pleistocene for Clade I, 1.2 MYA (0.7–1.8); late Pliocene for Clade II, 3.4 MYA (2.4–4.8); and mid-Pleistocene for Clade III, 1.0 MYA (0.3–1.9).

Table 1 Divergence time estimates for the Hynobius kimurae–H. boulengeri species complex.

Node	Cladogenetic event	Median	95% HPD interval	
1	Basal split of H. kimurae–H. boulengeri group (clades I+II vs. III)	7.0	(5.0–9.3)	
2	Split between Eastern (clade I) and Western (clade II) groups of H. kimurae s.lato	5.2	(3.7–6.9)	
3	Basal split of clade I (H. kimurae Eastern group)	1.2	(0.7–1.8)	
4	Basal split of clade II (H. kimurae Western group)	3.4	(2.4–4.8)	
5	Split between subclades II-1 and II-2	2.3	(1.6–3.3)	
6	Basal split of subclade II-1	1.5	(0.9–2.2)	
7	—	0.4	(0.2–0.7)	
8	Basal split of subclade II-2	1.1	(0.2–1.0)	
9	—	0.5	(0.5–2.1)	
10	—	1.4	(0.7–2.1)	
11	Basal split of clade III (H. boulengeri)	1.0	(0.3–1.9)	
12	Basal split of subclade I-1	0.7	(0.4–1.2)	
13	Basal split of subclade I-2	0.9	(0.5–1.4)	
14	Basal split of subclade II-3	0.3	(0.1–0.7)	
Note:

For node names see Fig. 4. Estimated age (median value and 95% highest posterior density (HPD) interval) is given in millions of years (MYA).

Figure 4 BEAST chronogram for the Hynobius kimurae—H. boulengeri species complex based on the mtDNA dataset (A).

Node values correspond to estimated divergence times (in Ma); blue bars reflect 95%-credibility intervals. Open circles on nodes a–c correspond to calibration points in BEAST analysis (see ‘Materials and methods’ for details). Closed circles on nodes 1–14 correspond to nodes with divergence time estimates shown in Table 1. The color of clades corresponds to the color of mitochondrial lineages in Figs. 1–3. Inset shows hypothetical paleogeographic reconstruction for Japanese islands during late Miocene (B; ca. 7.0 MYA), early Pliocene (C; ca. 5.0 MYA) and middle Pleistocene (D; ca. 1.5–1.0 MYA). Gray shading indicates land areas not submerged by sea. Red dotted lines indicate main tectonic trenches; red triangles indicate areas of active volcanism. Paleogeographic reconstructions are based on Minato, Gorai & Hunahasi (1965), Otofuji, Matsuda & Nohda (1985), Seno, Stein & Gripp (1993), Ninomiya et al. (2014) and Tojo et al. (2017).

Morphological variation

Among the three groups examined (“Western group,” H. kimurae sensu stricto from Kumogahata, Kyoto, population 19; “Eastern group,” H. kimurae sensu lato from Mt. Hinode, Tokyo, population six; and H. boulengeri), mean SVL varied significantly, ranging from 63.0 ± 2.4 mm (59.1–67.5 mm) in H. kimurae sensu stricto to 90.5 ± 1.5 mm (88.8–92.7 mm) in H. boulengeri (Table S5). Most morphological comparisons between H. kimurae and H. boulengeri were not significant due to small sample size of H. boulengeri (N = 3; see Table S5). Males of the Eastern group (SVL 74.6 ± 3.5 mm; 66.0–80.9 mm; df = 2; P = 0.0009) were significantly larger than males of the Western group, and were smaller than those of H. boulengeri. Furthermore, significant differences were found in comparisons of a number of character ratios and teeth number between the Eastern and Western group of H. kimurae (see Table S5, File S5).

In the PCA with both sexes analyzed together, PC Factor 1 explained 25.21% of the variability and PC Factor 2 explained 22.58% (Fig. 5). Contribution of morphological characters to the variables created by the PCA (in %) is shown in Table S6. All character ratios proved to be informative for the discrimination of the studied samples; RMTAW (8.64%), LJTN (8.43%), and RIND (8.36%) showed the largest differences, followed by RVTW (7.65%), UJTN (6.56%), and RSL (6.23%). Despite the limited sample size of H. boulengeri, the two-dimensional plot of the first two principal components for both sexes demonstrates complete discrimination of the three groups: (I) Western group (H. kimurae sensu stricto), (II) Eastern group (H. kimurae sensu lato), and (III) H. boulengeri (Fig. 5).

Figure 5 Two-dimensional plot of first against second factors of PCA of 29 morphological characters for the Hynobius kimurae–H. boulengeri species complex specimens examined.

Circles correspond to males, and triangles to females. Icon color corresponds to colors in Figs. 1–4. Photos by H. Okamiya and N.A. Poyarkov.

In summary, in both character ratios and meristic characters, all three examined members of the H. kimurae–H. boulengeri species complex form clearly separated morphological groups, with many characters significantly separating them from each other. H. kimurae sensu lato from the Eastern group is clearly morphologically distinct from H. kimurae sensu stricto and H. boulengeri (see Comparisons and “Discussion” for details).

Taxonomic Description

Our work clearly indicates that the H. kimurae–H. boulengeri species complex consists of three highly divergent lineages that correspond to three geographic groups of populations: Clade I in the montane areas of western Kanto and eastern Chubu District, Clade II in the mountains of central and northern Chubu, and westwards to Kinki and Chugoku Districts and Clade III restricted to Kii Mountains in the southern part of Kinki District (see Fig. 1). To our knowledge, these groups of populations have parapatric mode of distribution: no cases of sympatry between H. kimurae and H. boulengeri have been reported so far. We also did not find sympatric occurrence of mtDNA haplotypes of the Western and Eastern groups of H. kimurae in Chubu District. This corresponds with the results of an earlier electrophoretic study (Matsui et al., 2000).

The three groups of the H. kimurae–H. boulengeri species complex differ both in mtDNA sequences, nuDNA RAG1 gene sequences, and in external morphology. Genetic divergence between these lineages varies from 1.69% to 2.96% for16S rRNA gene and from 5.98% to 6.65% for cyt b gene. These divergence values are comparable or greater than those generally observed between closely related species in lotic Hynobius (see Lai & Lue, 2008; Nishikawa & Matsui, 2014; Matsui, Nishikawa & Tominaga, 2017) or in other groups of Hynobiidae (see Poyarkov & Kuzmin, 2008; Poyarkov et al., 2012; Xia et al., 2012; Min et al., 2016).

According to our molecular dating analysis, divergence between the three clades of the H. kimurae–H. boulengeri species complex likely took place within a comparatively short time frame during the late Miocene—early Pliocene. MtDNA genealogy suggests that H. boulengeri is a sister group to H. kimurae sensu lato, although with moderate nodal support. However, nuDNA analysis suggests that RAG1 haplotypes of the Eastern H. kimurae sensu lato are more divergent from H. kimurae sensu stricto than H. boulengeri. This indicates a possible discordance between the phylogenetic signals of mitochondrial and nuclear genomes. However, further studies involving additional nuDNA-markers are needed to address this problem. Nevertheless, the observed significant divergence between RAG1 haplotypes of the Western and Eastern H. kimurae groups suggests the presence of long-termed isolation between these lineages. These results are highly concordant with previous data of Matsui et al. (2000), who, using the electrophoretic analysis of 23 allozyme loci, clearly demonstrated that populations of H. kimurae are subdivided into two highly divergent groups, separated by mean Nei’s D of 0.497 (0.264–0.658). These two groups correspond to our Western and Eastern groups of H. kimurae sensu lato. A detailed morphological study by Matsui, Misawa & Nishikawa (2009) also showed that 24 populations from all over the H. kimurae sensu lato range in multivariate analysis of 24 morphological characters were divided into two major clusters that correspond to the Eastern and Western groups of H. kimurae discussed in the present paper.

Thus, concordant evidence from molecular analyses (mtDNA and nuDNA),external morphology and allozymes clearly indicate that the populations of H. kimurae sensu lato from the eastern part of its range represent a distinct new species of Hynobius, which we describe herein as: Hynobius fossigenus sp. nov.

(Figs. 6–10; Fig. S2; Tables S5 and S7–S9)

Synonymy:

Hynobius luteopunctatus (partim) Hatta (1914): 32; Hatta (1921): 654. Locality—“Honshu.” Nomen nudum. Synonymy according to Sato (1943): 218–234.

Pseudosalamandra kimurai (partim)—Tago (1931): 181.

Hynobius kimurai (partim)—Sato (1943): 218–234.

Hynobius (Hynobius) naevius kimurai (partim)—Nakamura & Ueno (1963): 12

Hynobius kimurae (partim)—Matsui in Sengoku (1979): 106–107; Matsui et al. (2000): 115–125; Matsui, Misawa & Nishikawa (2009): 87–95.

Holotype. ZMMU A-5862 (field number HN-17), adult male collected from a small stream on the eastern slope of Hinode-san (= Mt. Hinode), in the vicinity of Hinode City, Tokyo, Honshu Island, Japan (N 35.78°; E 139.17°; elevation 679 m a.s.l.), on March 1, 2018 by H. Okamiya.

Paratypes. With the exception of the holotype, the type series consists of 25 adult specimens: 17 males and eight females, all collected during the breeding season.

Type series includes: three adult males (ZMMU A-5863–A-5865; field numbers HN-18–20) with the same collection data as the holotype; five adult males (YCM-RA-581–585; field numbers: HN-01–05) and three adult gravid females (YCM-RA-581–585; field numbers: HN-10–12) collected from a small stream on the eastern slope of Hinode-san (= Mt. Hinode), in the vicinity of Hinode City, Tokyo, Honshu Island, Japan (N 35.78°; E 139.17°; elevation 679 m a.s.l.), on December 6, 2017 by H. Okamiya; four adult males (ZMMU A-5851–A-5854; field numbers: HN-06–09) and four adult gravid females (ZMMU A-5858–A-5861; field numbers: HN-12–16) collected from a small stream on the eastern slope of Hinode-san (= Mt. Hinode), in the vicinity of Hinode City, Tokyo, Honshu Island, Japan (N 35.78°; E 139.17°; elevation 679 m a.s.l.), on December 6, 2017 by H. Okamiya; five adult males (ZMMU A-5867–A-5871; field numbers: NAP-05241–05245) and one adult gravid female (ZMMU A-5866; field number: NAP-05240) collected from a small stream on the eastern slope of Hinode-san (= Mt. Hinode), in the vicinity of Hinode City, Tokyo, Honshu Island, Japan (N 35.78°; E 139.17°; elevation 679 m a.s.l.), on December 3, 2014 by N.A. Poyarkov, Y. Kuwabara, and T.Th. Kusunoki.

Referred specimens. A total of three adult males (ZMMU A-5885–A-5886, ZMMU A-5889; field numbers UR-001–002, UR-005) and three adult gravid females (ZMMU A-5887–A-5888, ZMMU A-5890; field numbers UR-003–004, UR-006) collected from a small stream in the vicinity of Urayama, Chichibu City, Saitama Prefecture, Honshu Island, Japan (N 35.90°; E 139.10°; elevation 638 m a.s.l.), on December 16, 2017 by K. Matsumoto; five adult males (ZMMU A-5891–A-5895; field numbers YMa-001–005) and three adult gravid females (ZMMU A-5896–A-5898; field numbers YMa-006–008) collected from a small stream in the vicinity of Kamisano, Nanbu City, Yamanashi Prefecture, Honshu Island, Japan (N 35.33°; E 138.51°; elevation 543 m a.s.l.), on December 29, 2017 by H. Okamiya; adult male (NSMT 1056) from Mt. Gozen-yama, Okutama City, Tokyo, Honshu Island, Japan, collected on June 18, 1960 by K. Tanaka; adult male (NSMT 3676) from Okusawa-dani, Hayakawa-cho, Minamikoma-gun, Yamanashi Prefecture, Honshu Island, Japan, collected on October 29, 1972 by M. Ogihara; two subadults (ZMMU A-5883–A-5884; field numbers NAP-07476–07477) collected from a small stream in the vicinity of Ootaba, Okutama City, Tokyo, Honshu Island, Japan (N 35.85°; E 139.15°; elevation 399 m a.s.l.), on January 28, 2007 by N.A. Poyarkov; two larvae (ZMMU A-5872–A-5873; field numbers YN-A24–A30) collected from a small stream in the vicinity of Kamisano, Nanbu City, Yamanashi Prefecture, Honshu Island, Japan (N 35.33°; E 138.51°; elevation 543 m a.s.l.), on December 29, 2017 by H. Okamiya; three larvae (ZMMU A-5874–A-5876; field numbers AC-A-3–A-5) collected from a small stream in the vicinity of Misawa, Toyone City, Aichi Prefecture, Honshu Island, Japan (N 35.22°; E 137.72°; elevation 958 m a.s.l.), on December 31, 2017 by H. Okamiya; two larvae (ZMMU A-5877–A-5878; field numbers TO-E-1–E-2) collected from a small stream in the vicinity of Kurakake, Hinohara City, Tokyo, Honshu Island, Japan (N 35.74°; E 139.05°; elevation 810 m a.s.l.), on December 7, 2017 by H. Okamiya; two pairs of egg sacs (ZMMU A-5879–A-5882; field numbers HN-ES-1–ES-4) with the same collection data as the holotype.

Diagnosis. The species is assigned to the genus Hynobius based on the following character states considered to be diagnostic for the genus: (1) lungs present; (2) digits in adults lack cornified claw-like structures; (3) dermal flaps on posterior edges of hindlimbs in adults absent; (4) tail relatively short, shorter than body, and distinctly flattened for most of its length; (5) vomerine teeth arranged in a distinctly curved series with inner branches notably longer than outer branches and located posterior to the level of choanae; (6) frontal fontanelle between frontals and parietals absent; (7) light, broad dorsal stripe absent; (8) fifth toe well-developed or absent (Dunn, 1923a, 1923b; Sato, 1943; Zhao et al., 1988; Adler & Zhao, 1990; Fei et al., 2006). In the external morphology Hynobius fossigenus sp. nov. most closely resembles the other Japanese lotic species of Hynobius, but is distinguished from its congeners by a combination of the following morphological attributes: (1) large-sized species (adult SVL 66.0–80.9 mm in males, 76.2–82.5 mm in females); (2) lotic-breeding Hynobius species (breeding in the mountain streams) with a cylindrical tail and a small number of large, unpigmented ova per clutch; (3) egg sacs thick, C- or G-shaped, with strong smooth iridescent envelope, forming a characteristic whiptail-like structure at the free end; (4) head comparatively small, distinctly longer than wide (in males HL/SVL ratio 23.2–25.2%; HW/SVL ratio 15.3–18.1%); (5) limbs slender and comparatively long; limbs adpressed to trunk separated by two to one half of costal fold (LON −2 to −0.5); (6) trunk slender (TRL/SVL ratio 74.5–79.8% in males; 77.0–79.9% in females) with usually 13 (occasionally 12) costal grooves; (7) comparatively long cylindrical tail compressed posteriorly (TL/SVL ratio 71.6–85.0% in males; 69.6–76.0% in females), with a low tail fin present in posterior one-fourth of its length; (8) fifth toe usually well-developed but may be absent; (9) vomerine teeth in two comparatively deep series forming a wide “U”-shaped figure, which is notably wider than long (VTW/VTL ratio 112.6–133.3% in both sexes), bearing 43–66 vomerine teeth (for both sexes); (10) upper jaws with 71–88 teeth, lower jaws with 54–81 teeth (for both sexes); (11) in life, dorsum dark purple-brown to blackish scattered with golden-yellow light spots, the number of which may vary from few to multitudinous, but they never form continuous light marbling pattern; underside lighter than dorsum, purplish-gray lacking light spots; iris dark brown without markings.

Description of holotype. An adult male, in a good state of preservation, initially fixed in 4% formalin and preserved in 70% ethanol, with an SVL of 69.42 mm (measured on the preserved specimen) (see Fig. 6).

Figure 6 Holotype of Hynobius fossigenus sp. nov. (ZMMU A-5862, male) in life.

(A) Dorsal view; (B) ventral view; (C) head, dorsal view; (D) head, ventral view; (E) head, lateral view; (F) opisthenar view of the right hand; (G) volar view of the right hand; (H) opisthenar view of the right foot; (I) plantar view of the right foot; (J) ventral view of cloacal area. Photos by N.A. Poyarkov.

Trunk. Head and body large; trunk slender and cylindrical (Figs. 6A and 6B); chest comparatively narrow: the CW/SVL ratio 14.2%. Skin on dorsum and venter smooth; diffuse microscopic glands scattered all over the body, well-notable on ventral surfaces. Mid-dorsal groove weakly developed, almost indistinct, extending from the basis of the head to the base of the tail (Fig. 6A). Costal grooves well developed, 13 grooves visible on each side of the body, 11 grooves visible from ventral side (Fig. 6B).

Cloaca. Cloaca slightly swollen, not protuberant in ventral and lateral views. Vent longitudinal, elongates dagger-shaped with slightly swollen edges and a small genital tubercle on anterior edge of cloacal slit (Fig. 6J).

Tail. Tail comparatively long, slightly shorter than body, TAL/SVL ratio 84.1%. Tail subcylindrical to oval in cross-section in the anterior half of its’ length, laterally compressed in the posterior half of its’ length; tail noticeably flattened in the posterior two-fifths of its length, oar-shaped with low dorsal tail fin evident posteriorly and rounded tail-tip. Tail width/height ratio at the middle of its length (MTW/MTH) 68.7%; the highest point of tail fin located on posterior one-fourth of its length. Tail widest near its base.

Extremities. Limbs slender and comparatively long (FLL/SVL 24.5%; HLL/SVL 29.1%), hindlimbs slightly longer and more robust than forelimbs (FLL/HLL84.3%) (Fig. 6); when forelimb and hindlimb adpressed against trunk, digit tips do not meet, separated by a gap equal to one costal segment (LON -1); forelimb length to axilla-groin distance ratio (FLL/AGD) 47.4%; hindlimb length to axilla-groin distance ratio (HLL/AGD) 56.3%. A total of two small rounded sore-like flattened palmar (metacarpal) tubercles located at the base of fingers I and IV; the external palmar tubercle slightly larger (Fig. 6G). Two flattened rounded metatarsal tubercles located at the base of toes I and V; the internal one slightly larger (Fig. 6I). Four fingers and five toes present; finger lengths in ascending order: I<IV<II<III (Figs. 6F and 6G); relative length of toes: I<V<II<IV<III (Figs. 6H and 6I). Fifth toe well-developed (5TL/SVL 5.1%). Digital webbing lacking; digits rounded in transverse section; digit tips rounded and slightly cornified (Figs. 6F–6I).

Head. Head oval and moderately depressed, comparatively small (HL/SVL ratio 25.2%), distinctly longer than wide (HL/HW ratio 143%); head basis slightly distinct from short neck (Figs. 6A and 6C). Tongue broad, convex-elliptical, adhering to mouth floor with free lateral margins. Snout wide and rather short (SL/HL ratio 25.3%); snout tip rounded in dorsal (Fig. 6C) and lateral views (Fig. 6E). Snout almost not projecting beyond lower jaw. Nostrils small, rounded, with lateral orientation, not protuberant, quite widely separated (IND/HL ratio 25.1%), located slightly closer to snout tip than to eye (ON/SL ratio 58.9%). Eyes large (UEL/HL ratio 21.9%), slightly protuberant in lateral view (Fig. 6E); slightly inset from head edge in dorsal view (Fig. 6C), eye diameter notably lower than snout length (UEL/SL ratio 86.2%) and lower than the distance between external nares (UEL/IND ratio 87.2%). Eyes widely spaced, interorbital distance longer than upper eyelid length (UEL/IOD ratio 90.1%). Upper eyelids present, well developed; labial folds absent; gular fold distinct, curving slightly anteriorly (Fig. 6D). Parotoid glands prominent, swollen, extending from jaw angle to gular fold. Longitudinal supraparotoidal groove deep (Fig. 6E), beginning above jaw angle and extending posteriorly, gently curving ventrally at head basis and intersecting with gular fold at its end. Postorbital grooves distinct, branching posterior to jaw angle, one short and running down to lower jaw, the other long and running posteriorly to parotoid gland (Fig. 6E). Postorbital groove subequal to eye length. Head dorsal surface with two distinct slightly curving lines of neuromasts extending posteriorly from the area between external nares to anterior corners of eyes and then further posteriorly along the orbit margins toward postorbital area (Fig. 6C).

Teeth. Vomerine teeth in two long, wide, obliquely arched series VTS, nearly touching at midline posteriorly, and forming a comparatively deep and wide “U”-shaped figure with no noticeable gap between branches. VTS anterior margin located at the level of anterior margins of choanae. Outer VTS branch ca. three times shorter than inner VTS branch; outer VTS branch slightly curved, reaching the level of the inner edge of the internal nares (Fig. 7I); inner VTS branches almost straight along anterior three-fourths of their length, with recurved posterior ends. Left and right VTS in contact with each other with no gap between the medial ends of the inner branches of VTS. VTS distinctly wider than long (VTW/VTL ratio 113.3%).

Figure 7 Morphological comparison of the Hynobius kimurae–H. boulengeri species complex members.

Upper row—egg sac morphology members in situ: (A) H. boulengeri (Nara Prefecture) (whiptail-like structure already fallen apart; photo by N. Kawazoe); (B) H. kimurae sensu stricto (Kyoto) (photo by N.A. Poyarkov); (C) Hynobius fossigenus sp. nov. (Tokyo) (photo by H. Okamiya). Middle row—typical dorsal coloration patterns of adults in life: (D) male H. boulengeri (ZMMU A-5842) (photo by N.A. Poyarkov); (E) male H. kimurae s. str. (ZMMU A-5904) (photo by N.A. Poyarkov); (F) male paratype of Hynobius fossigenus sp. nov. (YCM-RA-584) (photo by H. Okamiya). Lower row—open mouth cavities showing the shape of vomerine tooth series: (G) male H. boulengeri (ZMMU A-5842); (H) male H. kimurae s. str. (ZMMU A-5903); (I) male holotype of Hynobius fossigenus sp. nov. (ZMMU A-5862). Scale bar indicates 3 mm. Drawings by N.A. Poyarkov.

Measurements and counts of the holotype. All measurements in mm: SVL: 69.4; HL: 17.5; HW: 12.2; MXHW: 13.2; LJL: 9.7; SL: 4.4; IND: 4.4; IOD: 4.2; UEW: 2.4; UEL: 3.8; OL: 2.2; AGD: 35.9; TRL: 52.9; TAL: 58.4; BTAW: 8.2; MTAW: 6.3; MXTAH: 10.0; MTAH: 9.2; FLL: 17.0; HLL: 20.2; 2FL: 3.1; 3FL: 3.2; 3TL: 4.6; 5TL: 1.5; VTW: 4.1; VTL: 3.6; UJTN: 73; LJTN: 70; VTN: 53 (27/26); TN: 5; CGN: 13; LON: −1; ICD: 6.1; CW: 9.9; ON: 2.6; NSD: 2.1; 1FL: 1.4; 4FL: 1.6; 1TL: 1.2; 2TL: 3.1; 4TL: 3.2; CSL: 5.3.

Color of the holotype in life. Coloration in life is shown in Figs. 6 and 8. In life, dorsum dark purplish-brown to blackish scattered with numerous irregular small golden-yellow to golden-orange spots. Light spots are smaller and more numerous on dorsal surfaces of head and anterior part of dorsum getting more scarce on dorsal surfaces of limbs and tail. Background color of fingers and toes and lateral sides of head somewhat lighter grayish-purple. Underside lighter than dorsum, purplish-gray lacking light spots, uniformly covered with tiny light-gray spots of skin glands. Vent bluish-gray. Iris dark brown without markings. Cornified parts on digits dark brown.

Figure 8 Male holotype of Hynobius fossigenus sp. nov. (ZMMU A-5862) in situ.

Photo by H. Okamiya.

Color of the holotype in preservative. After six months in alcohol, the general coloration pattern of the holotype did not change significantly; however, the dorsal light markings faded to beige-gray, pinkish tint of the underside vanished to light gray.

Variation. All individuals in the type series appear to be generally similar in morphology and body proportions; variation of the type series in the morphometric characters is shown in Table S7. All individuals were examined after they had been preserved in 70% ethanol. Most specimens of the type series have 13 costal grooves (excluding the axillary and inguinal groove); however, four males had 12 costal grooves (YCM-RA-583–584; ZMMU A-5852, A-5871). Most specimens had all five toes well-developed; however, two males (YCM-RA-582 and ZMMU A-5851) had only four toes on the right foot. In all examined specimens limbs and digits were relatively short and adpressed limbs never overlapped, leaving two to a half of intercostal folds uncovered. A total of two males (ZMMU A-5852, A-5863) and two females (ZMMU A-5859, A-5861) had clear signs of the tail regeneration. No sexual differences were observed in the number of teeth on upper and lower jaw, number of vomerine teeth and the shape of the VTS, which corresponds well to that described for the holotype. Significant sexual differences were found in nine morphometric characters: males have significantly smaller body size, relatively shorter and wider heads, shorter axilla-groin distance and trunk length, relatively longer, thicker (tail width at the tail base and in the middle) and higher tails; males also have relatively longer forelimbs (see “Results” and Table S7 for details). Morphological variation of H. kimurae sensu lato, including five populations of the Eastern group corresponding to Hynobius fossigenus sp. nov., is analyzed in detail by Matsui, Misawa & Nishikawa (2009).

Coloration. Variation of the dorsal coloration in the type series is shown in Fig. S2. The dorsal background color does not vary significantly, but significant variation is observed in the number and abundance of small yellow spots on dorsum. All specimens of the type series are assigned to coloration type 2 (fewer than 10 dorsal yellow spots per cm2) or type 3 (10 or more dorsal yellow spots per cm2) of Matsui, Misawa & Nishikawa (2009). Of the type series, the male YCM-RA-582 has the fewest number of yellow spots on the dorsum. According to data of Matsui, Misawa & Nishikawa (2009), other populations of Hynobius fossigenus sp. nov. (in Kanagawa, Tokyo and Saitama Prefectures) may also exhibit coloration type 1 (dorsal yellow spots at most several in number).

Secondary sexual characteristics. During the breeding period, males’ cloacal area is noticeably swollen, with longitudinal vent slit, dagger-shaped with slightly swollen edges and a small genital tubercle on the anterior edge of the cloacal slit (Fig. 6J). The cloacal area of females is less swollen and the vent is a simple longitudinal slit. During the reproduction, males usually have a higher tail fin and females have notably swollen bodies.

Eggs and clutch. The egg sac morphology of Hynobius fossigenus sp. nov. is shown in Fig. 9; measurements of the egg sacs are presented in Table S8. Egg sacs are C- or G-shaped; female attaches both egg sacs to a stone with two mucous stalks at the egg sac basis (Fig. 9, ESB). Egg sacs of Hynobius fossigenus sp. nov. have very thick and almost completely smooth envelopes with peculiar bright bluish-violet iridescence when placed in water (see Fig. 9). The distal free end of each egg sac is gradually getting thinner and abruptly recurve forming a characteristic whiptail-like structure (Fig. 9, WT), which may be quite long reaching a length of up to 4 cm. Eggs large (egg diameter (EGD) 4.9–5.6 mm), yellowish in color, with egg capsules clearly seen through the envelope of the egg sac, usually placed in one row in the distal and proximal ends of the egg sac, and in two rows in the middle part of the egg sac. Eggs are few in number, usually 12–16 per egg sac (see Table S8). Breeding biology of Hynobius fossigenus sp. nov. Tokyo population is described by Misawa & Matsui (1997).

Figure 9 Egg sacs and larval stage of Hynobius fossigenus sp. nov. in life.

(A) Egg sac pair ZMMU A-5879–5880; (B) egg sac pair ZMMU A-5881–5882; lateral (C), dorsal (D), and ventral (E) views of a larva at stage 57; ZMMU A-5877. Study sites: ESB, egg sac base; WT, whiptail-like structure. Scale bar indicates 10 mm. Photos by H. Okamiya.

Larval morphology and metamorphosis. External larval morphology at the stage 57 is shown in Fig. 9; measurements of larvae are presented in Table S9. Larvae typical for lotic Hynobius, ca. 21 mm and lacking the balancer organ at hatching. Grown larvae hide under small stones in slowly-flowing parts of the stream. Head large, dorso-ventrally flattened, body comparatively slender, laterally flattened. Digits in developed larvae with small cornified claw-like structures. Dorsal fin starts at the level of the middle of the dorsum and reaches its greatest height at the middle of the tail. Tail subequal to SVL (TAL/SVL ratio 0.86–0.97). Dorsal tail fin slightly concave, ventral tail fin straight; tail tip tapered. Dorsally ochre-brown with numerous small dark-brown spots and blotches. Ventrally pale-yellow, no spots.

Phylogenetic position. The new species is a member of the H. kimurae–H. boulengeri species complex and is reconstructed as a sister species to H. kimurae sensu stricto based on the analyses of 16S rRNA and cyt b partial sequences (see Fig. 2; Fig. S1). The observed level of genetic divergence between the new species and other lotic Hynobius corresponds to a species level of differentiation (Lai & Lue, 2008; Nishikawa & Matsui, 2014; Matsui, Nishikawa & Tominaga, 2017). The split between Hynobius fossigenus sp. nov. and its sister species is estimated as 5.2 MYA (3.7–6.9 MYA).

Biochemical differentiation. Allozyme study of Matsui et al. (2000) based on the analysis of 235 individuals of H. kimurae sensu lato for 23 presumptive loci, showed that 20 loci were polymorphic and H. kimurae populations were grouped into three main clusters—Eastern (corresponding to Hynobius fossigenus sp. nov.), Central and Western groups. According to their data, Nei’s genetic distances (D) between Hynobius fossigenus sp. nov. and H. kimurae sensu stricto varied from 0.264 to 0.658 (mean 0.497). In general, Nei’s D of 0.22 or greater are reported to occur among sister species of Japanese Hynobius (Matsui, 1987), thus the D level observed in H. kimurae corresponds to the species-level of differentiation. Based on Nei’s Distances, Matsui et al. (2000) suggested that the split between the Eastern (Hynobius fossigenus sp. nov.) and the Central groups (H. kimurae sensu stricto) took place around 3.7 MYA, which is slightly more recent than our estimate.

Chromosomes. The karyotype and the C-banding pattern of the Tokyo population of Hynobius fossigenus sp. nov. was described by Ikebe, Yamamoto & Kohno (1986) and Ikebe & Kohno (1991). According to their data, diploid karyotype of the new species has 2n = 58 chromosomes with nine pairs of large-sized meta- and submetacentric chromosomes (Nos. 1–9), three pairs of submetacentric medium-sized chromosomes (Nos. 10–12), one pair of metacentric medium-sized chromosomes (No. 13), six pairs of meta- and submetacentric small-sized chromosomes (Nos. 14–19) and 10 pairs of acrocentric small-sized microchromosomes (Nos. 20–29). This karyotype is different from the karyotype of H. kimurae from Chubu District, but more closely resembles the karyotype reported for H. boulengeri (Ikebe, Yamamoto & Kohno, 1986).

Etymology. The specific name “fossigenus” is a Latinized adjective in masculine gender, derived from the Latin words “fossa” (meaning “pit,” “hollow”) and “-genus” (meaning “born in”). The new name is given in reference of the new species’ distribution, which is located on the both sides of the Itoigawa–Shizuoka Tectonic Line, the western boundary of Fossa Magna, the major rift zone in central Honshu (see Fig. 1). It is likely that the active uplifting of Central Highlands in Fossa Magna area during the early Pliocene caused the split between the new species and its sister species, H. kimurae sensu stricto. The suggested vernacular name in English is “Japanese Rift Salamander”; the suggested common name in Japanese: “Higashi-hida Sanshouuo.”

Reproduction, habitats, and natural history. As many other members of lotic Hynobius, Hynobius fossigenus sp. nov. reproduces in the headwaters of small mountain streams with cold well-aerated water; seasonal water temperature does not exceed 20 °C. The streams preferred for breeding are usually less than 1.5 m in width with ca. 20–30 cm in depth (Misawa & Matsui, 1997). Streams used for breeding are typically located in evergreen forests composed mostly by Cryptomeria japonica or mixed forests (Fig. 10). Metamorphosed individuals are found under rotten logs, fallen leaves, stones, or debris near the breeding stream.

Figure 10 Natural breeding habitat of Hynobius fossigenus sp. nov. at the type locality (Hinode Mt., Hinode, Tokyo City, Japan).

The mountain stream (A) and its headwaters (B), where larvae (C) and egg sacs (D) were encountered. Photos by H. Okamiya.

Normally, adults start to gather around the stream headwaters during November, where they hide under large stones and rocks partly submerged underwater. Breeding usually starts during December and lasts until April with the peak in egg laying observed between early February and mid-March, with water temperature varying from 5.5 to 7.0 °C (Misawa & Matsui, 1997). Reproduction takes place underwater, in the hollows under stones and rocks.

Female lays a pair of mucous egg sacs with thick and strong translucent envelopes, attaching them to the underside of large stones or rocks, or on smaller stones near waterfalls. The egg sacs are normally laid from early February to early April at water temperature 5.5–6.5 °C. Males tend to stay in the streams longer, they can be often found under the same stone with the deposited egg-clutches, while females appear to leave streams earlier, soon after oviposition. Hatching takes place from April to mid-May and, according to Misawa & Matsui (1997) embryonic development takes around 60 days. During hatching, the distal end of the egg sac breaks apart and larvae can freely leave the egg sac.

Larval growth in the Tokyo population of the new species was studied by Misawa & Matsui (1997). Normally, larvae reach full maturity in July after which their growth slows down. Larvae consume amphipods, larval caddis flies, and mayflies, and also prey on each other. In the stream, larvae are relatively easy to find under the stones or in the fallen leaves on the bottom of the stream (Fig. 10). In contrast to H. kimurae sensu stricto, Hynobius fossigenus sp. nov. is known to have overwintering larvae: most larvae in the Tokyo population overwinter at the last developmental stages and undergo metamorphosis at the end of May—early June of the next year; larval cannibalism with older larvae feeding on the younger larvae was also reported (Misawa & Matsui, 1997). Metamorphosed salamanders feed on small invertebrates such as spiders, insects, and earthworms. Misawa & Matsui (1999) reported that minimum maturation age in males is five years in Tokyo population, while females take at least seven years to reach maturity.

Hynobius fossigenus sp. nov. is the only lotic Hynobius species that occurs in Kanto and the eastern part of Chubu districts; however in Akaishi Mountains of Shizuoka and Nagano Prefectures its range overlaps with H. katoi; synoptic occurrence of these species was reported (Matsui et al., 2004).

Distribution. Approximate range of the new species is shown in Fig. 1 in red. To date, Hynobius fossigenus sp. nov. has been recorded from Gunma, Saitama, Tokyo and Kanagawa Prefectures of Kanto District, and from Yamanashi, Shizuoka, Nagano and the northeastern part of Aichi Prefecture in Chubu District of central Honshu, the main island of Japan (Sato, 1943; Matsui, 1979, 1981; Matsui et al., 2000; Matsui, Misawa & Nishikawa, 2009; Biodiversity Center of Japan, 2010). For a review of distribution of the new species in Gunma Prefecture see Kanai (2007); in Tokyo City area see Kusano, Ueda & Hatsushiba (2001); in Kanagawa Prefecture see Yamazaki et al. (1997); in Yamanashi Prefecture see Ogihara & Nakamura (1974); in Aichi Prefecture see Ohtake, Sakakibara & Yamagami (2009). It appears that the range of Hynobius fossigenus sp. nov. consists of several (at least five) geographic populations restricted to the separate mountain massifs and most likely isolated from each other. The border between the known ranges of Hynobius fossigenus sp. nov. and H. kimurae sensu stricto coincides with Yahagi-gawa River valley in the northeastern part of Aichi Prefecture (Mino-Mikawa Kogen area), and the cases of sympatric occurrence of both species are unknown to date.

Hynobius fossigenus sp. nov. was recorded from elevations 300–1100 m a.s.l. and is most abundant at elevations ca. 400–900 m a.s.l. (according to Kusano, Ueda & Hatsushiba, 2001; Matsui et al., 2000; Matsui, Misawa & Nishikawa, 2009).

Comparisons. Among the 37 currently recognized species of the genus Hynobius (see Frost, 2018), H. turkestanicus Nikolskii appears to be an enigmatic taxon (see Kuzmin, 1999) and is likely not a member of this genus (Min et al., 2016). The rest of the species are assigned to three eco-morphological groups that differ in morphology, phylogenetic position, chromosome structure, breeding ecology, and natural history. The lentic-breeding (or pond-type) Hynobius found in Japan, Korea and central, eastern and northeastern China, as well as H. (Satobius) retardatus, found in Hokkaido, are markedly distinct from the lotic-breeding (stream-type) species of Hynobius that inhabit the montane areas of Taiwan and Japan, as they have laterally compressed tails and deposit a large number of small, pigmented ova per clutch (vs. tail cylindrical at the base and small number of large unpigmented eggs per clutch in lotic Hynobius species). Thus, comparisons of the new species with other lotic Hynobius are most pertinent.

Hynobius fossigenus sp. nov. can be distinguished from the Taiwanese congeners H. fucus Lai & Lue, 2008; H. formosanus Maki, 1922; H. glacialis Lai & Lue, 2008; H. arisanensis Maki, 1922 and H. sonani (Maki, 1922) by its dorsal coloration and much larger body size (SVL 66.0–82.5 mm (mean 75.7) in the new species vs. all Taiwanese Hynobius–they are much smaller with SVL never exceeding 69 mm).

Among the other lotic Hynobius species, Hynobius fossigenus sp. nov. is distinct in having egg sacs with strong almost smooth envelopes with bright bluish-violet iridescence when placed in water, with egg sac free end recurved and forming a whiptail-like structure (see Fig. 9). Such morphology of the egg sac envelopes is characteristic only for members of the H. kimurae–H. boulengeri species complex (see Figs. 7A–7C) and is not observed in other lotic Hynobius species of Japan or Taiwan (see Nishikawa, Sato & Matsui, 2008; Nishikawa & Matsui, 2014; Matsui, Nishikawa & Tominaga, 2017). By having dark blackish-violet dorsum with small yellow spots, Hynobius fossigenus sp. nov. can be distinguished from the species with completely dark dorsum lacking light markings, which were previously regarded under the name H. boulengeri (Nishikawa et al., 2007). These species include H. boulengeri sensu stricto from Kii Mountains in Honshu Island (Fig. 7D), H. hirosei from Shikoku Island, H. shinichisatoi, H. osumiensis and H. amakusaensis from Kyushu Island, and H. katoi from Akaishi Mountains in the central Honshu Island. All these species can be easily differentiated from the new species by having completely dark dorsum that lacks light markings, or nearly immaculate dark dorsum with occasional small whitish dots as in H. amakusaensis and H. katoi (see Matsui et al., 2004; Nishikawa & Matsui, 2014). Among them, H. katoi, which is sympatric with Hynobius fossigenus sp. nov., can be further distinguished from the new species by its much smaller body size (SVL 53.8–66.1 mm vs. 66.0–82.5 mm (mean 75.7) in the new species) and much shallower VTS with a lower number of vomerine teeth (VTN 39 in the holotype of H. katoi vs. VTN 43–66 (mean 55.8) in Hynobius fossigenus sp. nov.) (see Matsui et al., 2004).

Hynobius ikioi (until recently regarded as H. stejnegeri) from the mountains of central Kyushu can be distinguished from the new species by jet-black background color of the dorsum with clear orange yellow confluent markings that form marble-like pattern (vs. dark purplish-brown background color of the dorsum with small yellow spots in Hynobius fossigenus sp. nov.). Males of H. ikioi also have generally longer tail (RTAL 68.2–97.3, mean 85.4 vs. 71.6–85.0, mean 78.6 in the new species), longer third finger (R3FL 3.3–5.1, mean 4.4, vs. 2.7–4.6, mean 3.8 in the new species), longer third toe (R3TL 5.9–8.2, mean 7.0, vs. 5.8–6.8, mean 6.3 in the new species), and generally deeper VTS (VTW/VTL 85.1–138.5, mean 106.8, vs. 112.6–131.4, mean 119.5 in the new species) than Hynobius fossigenus sp. nov.(see Matsui, Nishikawa & Tominaga, 2017).

Hynobius naevius occurring in the montane areas of the western Honshu and the northern Kyushu Islands can be easily distinguished from the new species by dorsum coloration. This species has a body with bluish- or reddish-purple background color, and pale-white lateral markings (vs. dark blackish-violet dorsum with small yellow spots in Hynobius fossigenus sp. nov.). H. stejnegeri (until recently regarded as H. yatsui Oyama, 1947) occurring in the mountains of the southern, central and eastern Kyushu, Shikoku and central part of Honshu, can be differentiated from the new species by its body size, proportions and coloration. H. stejnegeri males have much smaller body size (SVL less than 64.5 mm, mean 59.3 mm vs. 66.0–80.9, mean 74.6 mm in the new species) (see Tominaga et al., 2005). Coloration of H. stejnegeri is quite variable, but this species’ dorsum background color is always reddish-purple with discontinuous whitish markings or white dots or continuous orange to reddish markings (vs. dark blackish-violet dorsum with small yellow spots in Hynobius fossigenus sp. nov.) and can be readily distinguished from the new species (see Tominaga et al., 2005).

Among the members of the H. kimurae–H. boulengeri species complex, Hynobius fossigenus sp. nov. is most easily distinguished from H. boulengeri which has completely uniform dark bluish-black dorsum that lacks light markings (Fig. 7D) (vs. dark blackish-violet dorsum with small yellow spots in Hynobius fossigenus sp. nov.; Fig. 7F). Males of the new species can be further distinguished from the males of H. boulengeri by having a comparatively narrower head (RHW 15.3–18.1, mean 16.6, vs. 18.7–20.2, mean 19.5, in H. boulengeri; RMXHW 16.3–19.6, mean 17.8, vs. 20.3–20.9, mean 17.8, in H. boulengeri), comparatively shorter lower jaw (RLJL 11.4–14.6, mean 12.9 vs. 14.3–15.4, mean 15.0 in H. boulengeri), and a shorter snout (RSL 6.3–7.4, mean 6.7, vs. 7.4–7.6, mean 7.5 in H. boulengeri) (see Table S5 for details).

Finally, from its sister species, H. kimurae sensu stricto, the new species can be distinguished by the following combination of morphological attributes (see Table S5 for details; H. kimurae sensu stricto specimens used for comparison originated from Kumogahata, Kyoto, ca. 12 km from the type locality of H. kimurae at Mt. Hieizan, and share the same mtDNA and nuDNA haplotypes as the topotype specimens, see File S2). Both sexes of Hynobius fossigenus sp. nov. have larger body size, male SVL mean 74.6 mm (66.0–80.9 mm; N = 18) in the new species vs. mean 63.0 mm (59.1–67.5 mm; N = 9) in H. kimurae sensu stricto; female SVL mean 78.4 mm (76.2–82.5 mm; N = 8) in the new species vs. mean 71.2 mm (67.4–74.9 mm; N = 3) in H. kimurae sensu stricto. Furthermore, males of the new species have a comparatively shorter head, RHL mean 23.9 (23.2–25.2) (vs. RHL mean 25.6 (23.9–26.9) in H. kimurae sensu stricto). Males of Hynobius fossigenus sp. nov. have comparatively greater distance between external nares, RIND mean 6.5 (5.9–7.2) (vs. RIND 5.6 (5.0–6.6) in H. kimurae sensu stricto). Males of the new species have comparatively smaller upper eyelid width (RUEW mean 3.3 (3.2–3.6) in the new species vs. mean 3.5 (3.4–3.7) in H. kimurae sensu stricto) and smaller orbit length (ROL mean 2.9 (2.6–3.2) in the new species vs. mean 3.5 (3.0–3.9) in H. kimurae sensu stricto). Males of Hynobius fossigenus sp. nov. have also comparatively longer trunks, RTRL mean 77.2 (74.5–79.8) (vs. RTRL mean 74.3 (73.5–75.3) in H. kimurae sensu stricto). Males of Hynobius fossigenus sp. nov. have comparatively longer tails, though this character is affected by the tail regeneration after damage; in males of the new species RTAL mean 78.6 (71.6–85.0) vs. mean 70.7 (68.0–75.1) in H. kimurae sensu stricto. Furthermore, males of the new species have much thicker tails, with RMTAW mean 9.0 (7.8–10.0) (vs. comparatively more thin tails with RMTAW mean 7.7 (6.4–8.6) in males of H. kimurae sensu stricto).

In addition to the differences in coloration and body proportions, Hynobius fossigenus sp. nov. can be easily distinguished from H. kimurae sensu stricto by the shape of the VTS. The new species has comparatively shallow, U-shaped VTS (see Fig. 7I), which are wider than long (VTW/VTL 112.6–131.4, mean 119.5 for males; see Table S5) and have more resemblance with the shape of VTS in H. boulengeri (see Fig. 7G), whereas H. kimurae sensu stricto normally has long, deep, V-shaped VTS (see Fig. 7H), which are always longer than wide (VTW/VTL 75.6–80.9, mean 78.1 for males; see Table S5). Finally, Hynobius fossigenus sp. nov. has significantly higher number of teeth in both upper and lower jaws: UJTN 71–88 (mean 77.5), LJTN 60–81 (mean 68.4) in the new species vs. UJTN 49–57 (mean 52.9), LJTN 38–47 (mean 42.6) in H. kimurae sensu stricto.

Conservation status. At present, Hynobius fossigenus sp. nov. is not protected on international or national level, but is protected in Gunma prefecture as H. kimurae (Kanai, 2012). The new species is locally quite abundant. However, some isolated populations of the new species may be affected by the anthropogenic influence and habitat destruction. Given the available information, we suggest that Hynobius fossigenus sp. nov. be considered least concern (LC), following IUCN’s Red List categories (IUCN, 2001).

Discussion

Our study provides an assessment of the molecular variation within the H. kimurae–H. boulengeri species complex for the first time. Our data confirms the monophyly of the complex and its position in the genealogical tree of the genus as a sister lineage to all the other Hynobius species except H. (Satobius) retardatus. Thus, the lotic species of Hynobius, previously regarded as a separate genus or subgenus (see Tago, 1931; Nakamura & Ueno, 1963) are divided into at least four clades distantly related to each other: (1) H. kimurae–H. boulengeri species complex; (2) Taiwanese Hynobius; (3) Kyushu lotic Hynobius (H. ikioi and related species); and (4) H. naevius–H. stejnegeri species complex (Fig. S1). Interestingly, H. hidamontanus Matsui, 1987 from Hakuba in Chubu District, which is traditionally regarded as a lentic-breeding species, is nested deeply in the group of lotic Hynobius together with H. naevius, H. stejnegeri, H. katoi and H. hirosei (Fig. S1). It is noteworthy that H. hidamontanus was reported to lay the eggs under stones in small streams (Hasumi, Kakegawa & Saikawa, 2002), indicating that the reproduction mode in this species may be somewhat intermediate between the lotic and lentic breeding types. This data suggests that the ecological shifts between the stream-breeding and still water-breeding biology might have occurred several times in the evolutionary history of Hynobius salamanders. However, the phylogenetic position of the H. kimurae–H. boulengeri species complex as sister to the majority of other Hynobius implies that the stream-breeding adaptations were developed at the early stages of genus differentiation.

Our results go in line with the previous works which also indicated sister-species relationships of H. kimurae s. lato and H. boulengeri (Larson, Weisrock & Kozak, 2003; Li, Fu & Lei, 2011; Zheng et al., 2012; Nishikawa & Matsui, 2014), with the exception of Weisrock et al. (2013), who suggested that H. kimurae and H. hirosei form a clade (see File S1 for details). Despite the different morphology and coloration patterns, H. kimurae and H. boulengeri were found to be surprisingly closely related to each other, and the RAG1 haplotype network even suggests the paraphyly of H. kimurae sensu lato with H. boulengeri. The main character which is unique for the members of the H. kimurae–H. boulengeri species complex is the specific morphology of the egg sac with very strong iridescent envelopes forming whiptail-like structure at the free end. It is likely that this specific morphology of the egg sac envelope has developed as an adaptation toward egg-laying in the montane streams with strong current or near waterfalls, and it represents a synapomorphy of the H. kimurae–H. boulengeri species complex.

According to our knowledge, members of the H. kimurae–H. boulengeri species complex have parapatric distributions (see Fig. 1). H. boulengeri is restricted to Kii Mountains in Wakayama, Nara and Mie prefectures. The border that separates the ranges of Hynobius fossigenus sp. nov. and H. kimurae sensu stricto is located in the northern part of Aichi Prefecture and continues to the southern part of Nagano Prefecture; it likely coincides with the Yahagi-gawa River valley. Interestingly, based on the results from the allozyme study, Matsui et al. (2000) recorded only one form of H. kimurae in Aichi Prefecture, corresponding to Hynobius fossigenus sp. nov. (population five from Inabu-cho, Aichi Pref.), while our study reports the occurrence of both species in the northern part of Aichi Prefecture (see Fig. 1). Our study also indicates the significant divergence between the Ishikawa population of H. kimurae sensu stricto and the main part of its range in Kinki and Chubu Districts both in mtDNA and nuDNA markers; further studies with denser sampling are required to elucidate phylogeographic pattern within H. kimurae sensu stricto.

In general, the geographic split between Hynobius fossigenus sp. nov. and H. kimurae sensu stricto corresponds to the border between the eastern and western parts of Japan and a similar phylogeographic pattern was reported in various groups of animals, including freshwater fishes (Watanabe, 1998, 2012; Watanabe et al., 2006, 2014; Tominaga, Nakajima & Watanabe, 2016; Nakagawa et al., 2016) and insects (Suzuki, Sato & Ohba, 2002; Tojo et al., 2017). This initial split is often attributed to the formation of the Fossa Magna, which separated the southwestern and northeastern paleo-Japanese landmasses (Kato, 1992). The two landmasses gradually connected after 15.0 MYA, followed by the uplifting of Central Highland mountain ranges during the Late Miocene–Early Pliocene (Machida et al., 2006). Tectonic movements involving the uplift of mountains and volcanism became active especially in the Pleistocene, resulting in mountain coverage of over than 60% of the archipelago (Tominaga, Nakajima & Watanabe, 2016).

The speciation within the H. kimurae–H. boulengeri species complex was likely driven by the events during the formation of the Honshu mainland; the hypothetical paleogeographic scenario is shown in Fig. 4. The initial radiation within the species complex happened around 7.0 MYA (5.0–9.3 MYA) and likely took place after the formation of Fossa Magna and the initial separation of the southwestern and northeastern paleo-Japanese landmasses. According to our scenario, the ancestor of the H. kimurae–H. boulengeri species complex likely in habited the southwestern paleo-Japanese landmass (Fig. 4A). It is unclear what led to the separation of Clade III from the ancestors of Clades I+II, but this period coincides with intensive crust movement along the Japanese Median Tectonic Line (JMTL, see Figs. 1 and 4), which started around 7.0 MYA and led to formation of the Seto Inland Sea and production of uneven landforms all over the western Japan (Kosaka, 1995; Chinzei & Machida, 2001). Presently, the range of H. boulengeri lies mostly southwards from the JMTL, while H. kimurae is distributed northwards from this line (see Fig. 1).

After the closure of the Fossa Magna channel, the common ancestor of Clades I+II could disperse on the northeastern landmass connected to the southwestern landmass (Fig. 4B). We estimated the time of the separation of the Hynobius fossigenus sp. nov. and H. kimurae sensu stricto ancestors to be 5.2 MYA (3.7–6.9 MYA). This falls into the early Pliocene and nearly coincides with the mid-Pliocene times when the Japan mainland ceased the clock-wise rotation and settled at its present position (Otofuji & Matsuda, 1984). In fresh water organisms, the differentiation of the sister species or subspecies across the center of Honshu is often hypothesized to result from the vicariance associated with uplift of the Central Highlands in Pliocene (ca. 3.0–6.0 MYA; Machida et al., 2006; Watanabe et al., 2006; Tominaga, Nakajima & Watanabe, 2016). This period also coincides with intensive volcanism in Central Highlands and formation of Hida Mountains at the approximate location where the boundary between the present-day ranges of Hynobius fossigenus sp. nov. and H. kimurae sensu stricto lies (Kosaka et al., 1992; Kosaka, 1995; Takeuchi, 1999). This tectonic activity might have been responsible for the vicariant event, which separated the western and eastern lineages of H. kimurae sensu lato (see Fig. 4B).

The range of Hynobius fossigenus sp. nov. encompasses a number of isolated populations distributed across the Itoigawa-Shizuoka Tectonic Line (ISTL, the western edge of Fossa Magna, see Fig. 1) and forms an incomplete ring around the volcanic area of Mt. Fuji on the border of Shizuoka and Yamanashi Prefectures. Our analysis shows the separation of Hynobius fossigenus sp. nov. into two subclades I-1 and I-2, distributed eastwards and westwards from Mt. Fuji area, respectively. This split is estimated to occur1.2 MYA (0.7–1.8 MYA), which coincides with the collision of the Izu paleo-land with the Japanese mainland around 1.0 MYA, and the consequent intensified tectonic activity in Mt. Fuji area that reached the peak around 0.7 MYA (Kosaka et al., 1992; Kosaka, 1995; Takeuchi, 1999; Hisada, Ito & Togami, 2008; Niitsuma, 2007). Basal divergence within H. kimurae sensu stricto is much older and is estimated to occur 3.4 MYA (2.4–4.8 MYA); the factors that might have led to this divergence remain unclear. Further studies are needed to assess differentiation within H. kimurae sensu stricto in more detail.

Conclusion

In conclusion, our study clarified the phylogeographic pattern of the H. kimurae–H. boulengeri species complex and documented the presence of a new species—Hynobius fossigenus sp. nov., inhabiting central Honshu. Phylogenetic analyses of mtDNA supported the monophyly of the H. kimurae–H. boulengeri complex and suggested it as a sister lineage to all other Hynobius except H. (Satobius) retardatus. Among the other lotic Hynobius species, the members of the H. kimurae–H. boulengeri complex are characterized by the unique morphology of the egg sacs, which may serve as an adaptation toward egg-laying in streams with strong currents or near waterfalls.

The distribution processes of the montane stream-dwelling salamanders are generally thought to be affected by the geological factors, such as volcanism and mountain uplift, and repeated regressions and transgressions of sea level. Stream-breeding Hynobius serve as a promising model for the studies of historical biogeography during the formation of Japanese landmass. The basal split within the H. kimurae–H. boulengeri complex is estimated to occur in the late Miocene (ca. 7.0 MYA) and coincides with intensive crust movement in western Japan in the late Miocene, while divergence between Clades I and II took place in the early Pliocene (ca. 5.2 MYA) and was likely driven by the uplift of Central Japanese Highlands and intensive volcanism in the early Pliocene.

Our study demonstrates the importance of geological history of Honshu Island for the formation of diversity in freshwater fauna and illustrates that the taxonomy and phylogeography of Japanese Hynobius are still insufficiently studied. Phylogeographic history and profound genetic differentiation among regional populations of Hynobius fossigenus sp. nov. and H. kimurae sensu stricto suggest the necessity to protect them separately as independent management units. Our work calls for further studies of phylogeography of both Hynobius fossigenus sp. nov. and H. kimurae sensu stricto, as well as for examination of the populations across the border area between the two species.

Supplemental Information

Supplemental Information 1 Fig. S1. Phylogenetic BI tree of Hynobius kimurae sensu lato and related species based on 2,345 bp (partial 16S rRNA and cyt b sequences) with enlarged sampling of lotic Hynobius species.

Values on the branches correspond to BI PP/ML BS, respectively; filled, grey and white circles correspond to well-supported, moderately supported and non-supported nodes, respectively. Red color denotes lotic (stream-breeding) Hynobius, black color denotes lentic (still water-breeding) Hynobius species. Hynobius (Satobius) retardatus is a sister taxon to all other Hynobius and combines features of both lotic and lentic species. For locality information and voucher info see Tables S1 and S3.

Click here for additional data file.

Supplemental Information 2 Fig. S2. Variation of dorsal coloration in the type series of Hynobius fossigenus sp. nov.

Upper row and three specimens from the right in the lower row–males; seven specimens from the left in the lower row–females. Scale bar indicates 10 mm. Photos by H. Okamiya and N.A. Poyarkov.

Click here for additional data file.

Supplemental Information 3 Table S1. Specimens and sequences of Hynobius kimurae sensu lato and related Hynobius species used for molecular analyses.

For details on geographic location of examined populations see Table S2.

Click here for additional data file.

Supplemental Information 4 Table S2. Locality information for specimens of the Hynobius kimurae–H. boulengeri species complex used in molecular analyses.

Population numbers and names correspond to those in Table S1 and Fig. 1.

Click here for additional data file.

Supplemental Information 5 Table S3. Additional cyt b sequences of Hynobius species used for molecular analyses.

Click here for additional data file.

Supplemental Information 6 Table S4. Genetic divergence of the Hynobius kimurae–H. boulengeri species complex.

Uncorrected p-distances (percentage) for 16S rRNA (below diagonal) and cyt b (above diagonal) sequences of the Hynobius kimurae–H. boulengeri species complex members and other Hynobius species included in phylogenetic analyses. Within-clade genetic distances for 16S rRNA/cyt b genes are shown on the diagonal in bold.

Click here for additional data file.

Supplemental Information 7 Table S5. Means ± SD of SVL (in mm), medians of metric character ratios (% SVL) and meristic characters for males of the Hynobius kimurae–H. boulengeri species complex members.

Ranges are shown in parentheses. For character abbreviations, refer to Supplemental Information 4. N–sample size. Star (*) indicates that Tukey-Kramer test (for SVL) or Kruskal-Wallis test (other characters) show significant differences (p<0.05) for comparisons between Hynobius fossigenus sp. nov. and H. kimurae s. str.; double star (**) indicates Kruskal-Wallis test showing significant differences (p<0.05) for comparisons between Hynobius fossigenus sp. nov. and H. boulengeri.

Click here for additional data file.

Supplemental Information 8 Table S6. Character ratio contributions to factors 1, 2 and 3 of principal component analysis (percentage).

Click here for additional data file.

Supplemental Information 9 Table S7. Measurement data for Hynobius fossigenus sp. nov. type series.

For character abbreviations see Supplemental Information 4. 1–Character or character ratio to SVL exhibits sexual dimorphism (see Materials and methods; p<0.05); 2–denotes regenerated tail. Paired meristic characters (TN, VTN) are given in right/left order. Male holotype (ZMMU A-5862) marked in bold and with star (*).

Click here for additional data file.

Supplemental Information 10 Table S8. Measurement data for the egg sacs of Hynobius fossigenus sp. nov.

For character abbreviations see Supplemental Information 4.

Click here for additional data file.

Supplemental Information 11 Table S9. Measurement data for larvae of Hynobius fossigenus sp. nov.

For character abbreviations see Supplemental Information 4. Staging is estimated according to Akita (2001).

Click here for additional data file.

Supplemental Information 12 Supplemental Text Files.

Click here for additional data file.

HO thanks Kaoru Matsumoto (Saitama) for assistance in the field. HO also thanks Ryota Matsuyama (Tokyo) and Rena Nakajima (Kanagawa) for the support surveys and assistance with experiments. NAP thanks Evgeniya N. Solovyeva (Moscow) and Tang Van Duong (Moscow) for help with the data analysis; Koji Iizuka (Tokyo) and Masaki Kuro-o (Hirosaki) for support during his surveys in Japan. We are most grateful to Nobuhiro Kawazoe (Kyoto) and Yuusuke Kuwabara (Tokyo) for providing photos, sharing data and help in the field. NAP also thanks Tetsuya Thomas Kusunoki (Osaka/Moscow), Taku Shibata (Kyoto), Kazuma, Masahiro, Yukari and Yuuya Ishihara (Kani, Gifu) for constant help and support. NAP thanks Evgeniy S. Popov (St. Petersburg), Maxim S. Nuraliev (Moscow) and Sergei V. Kruskop for the fruitful discussions. We are deeply grateful to Graham Wallis (Otago) and the three anonymous reviewers for useful comments which helped us to improve the earlier version of the manuscript.

Additional Information and Declarations

Competing Interests

Author Contributions

Animal Ethics

Field Study Permissions

DNA Deposition

Data Availability

New Species Registration

The authors declare that they have no competing interests.

Hisanori Okamiya conceived and designed the experiments, performed the experiments, analyzed the data, contributed reagents/materials/analysis tools, prepared figures and/or tables, authored or reviewed drafts of the paper, approved the final draft, discussion of the results.

Hirotaka Sugawara conceived and designed the experiments, performed the experiments, contributed reagents/materials/analysis tools, authored or reviewed drafts of the paper, approved the final draft, discussion of the results.

Masahiro Nagano conceived and designed the experiments, analyzed the data, contributed reagents/materials/analysis tools, authored or reviewed drafts of the paper, approved the final draft, discussion of the results.

Nikolay A. Poyarkov conceived and designed the experiments, performed the experiments, analyzed the data, contributed reagents/materials/analysis tools, prepared figures and/or tables, authored or reviewed drafts of the paper, approved the final draft, discussion of the results.

The following information was supplied relating to ethical approvals (i.e., approving body and any reference numbers):

Specimen collection protocols and animal operations followed the Institutional Ethical Committee of Animal Experimentation of the Tokyo Metropolitan University (certificate number 20 of April 1, 2007) and strictly complied with the ethical conditions of the Science Council of Japan (June 1, 2006).

The following information was supplied relating to field study approvals (i.e., approving body and any reference numbers):

Field work, including collection of samples and animals in the field, was authorized by the education board of Utsunomiya, Tochigi, Japan (permit number 102 of April 22, 2014 issued to F. Hayashi and H. Sugawara).

The following information was supplied regarding the deposition of DNA sequences:

Sequences of 16S rRNA, cyt b, and RAG-1 genes presented here are accessible via GenBank accession numbers MH253618–MH253668 and MH287353–MH287433.

The following information was supplied regarding data availability:

Specimens examined in this study are deposited in herpetological collections of the following museums: Zoological Museum of Moscow University (ZMMU, Moscow, Russia);

Yokosuka City Museum (YCM, Yokosuka, Kanagawa Province, Japan).

The following information was supplied regarding the registration of a newly described species:

Publication LSID:

urn:lsid:zoobank.org:pub:AE462D10-3947-445D-8B3B-090675EDBA91;

Species name: Hynobius fossigenus,

urn:lsid:zoobank.org:act:B4AE334E-89BB-499F-AE05-C8BCB3658519.

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
