# Peer review of "An integrative taxonomic analysis reveals a new species of lotic Hynobius salamander from Japan"

_PeerJ, doi:10.7717/peerj.5084_

## Round 0.1 · original submission · Minor Revisions

Three reviewers have provided enormously helpful and detailed reviews of your manuscript. The main messages, with which I strongly agree, is that the paper is too long, detailed and verbose, and that the grammar requires attention in parts. There are 38 pages of text alone in a total of 93 pages!

I do have some sympathy, however, that this paper is essentially a formal taxonomic monograph with phylogeographic information embedded within- both the reviewers and I are probably more used to reading shorter, punchier phylogeography papers! I don't want to compromise the expectations of good taxonomic practice, so I make a general plea to reduce the length wherever you can. In particular, there should be no repetition, and there could be less background natural history etc. The writing could be less verbose throughout; use referencing wherever possible (e.g. description of morphological measurement, primers etc has probably been given elsewhere). The Abstract is far too long- 200-250 words should be ample. Finally, get someone to check grammar, particularly sing/pl and articles.

Please reduce datings to a single decimal place- Table 3 presents them to 4dp, i.e. the nearest century! Ratios are not normally used in morphological taxonomy any more (except perhaps for field ID). PCA handles absolute differences.

Another area where there can be a big reduction in length is in the Tables & Figures. All of the Tables, with the possible exception of Table 3, should be placed as Supplementary, for the rare reader that might want to access primary data. These sorts of details are no longer published with the main body of the paper. I suggest that Figs. 7, 9-11, 13-14 are only "supportive" of the main thrust/data of the paper, and should be Supplemental. That still leaves several Figs. of morphology and habitat of the new species. If a reader wants info on, for example, polytypism, they can look at the Supplementaries.

The reviewers make a number of other detailed suggestions which should be considered in production of the final ms. It is clear that much work has gone into this ms by both authors, and reviewers- it deserves careful amendment!

Reviewer 1 ·

Basic reporting

The interesting paper reveals the phylogenetic structure of a group of Hynobius salamanders from Japan, resulting in biogeographic inferences and the description of a new species. The study is novel and original, and the topic is attractive to PeerJ. The basic presentation of the manuscript however needs improvements before acceptance can be considered. Below are the main points that caught my eye:

1. While the language is of sufficient standard to enable comprehension of all lines of arguments, the entire text still needs to become polished for example by an experienced native speaker. Many sentences contain little grammatical errors and awkward phrases which are not picked up by a spell checker. One example is the frequent lack of the article “the” connected to respective nouns – see e.g. line 20, which should read “…..that cover almost the entire range of the H. kimurae – H. boulengeri complex….”, and not ““…..that cover almost entire range of H. kimurae – H. boulengeri complex….”. There are many similar passages, and generally many similar examples of phrasing which needs to be polished. Also be consistent with the spelling of important terms (e.g. “egg-sac” vs. “egg sac”; use the latter).

2. Parts of the manuscript are arguably too long, and not all information is necessary to convey the main take home messages of the work. The > 1000 lines of text (excl. references) are not fully warranted. Because so many taxon-specific details are presented, the current style of the manuscript often resembles a paper for example in Herpetological Monographs, and findings of general relevance are getting a little lost in between taxon-specific details. Aspects which could become shortened are the detailed taxonomic history of related Hynobius species (not everything is fully relevant for the present work), the natural history details linked to the species description (most of the presented facts are already published elsewhere), and somewhat repetitive parts in the Discussion and Conclusion sections.

3. The main lines of arguments throughout the Introduction and Discussion are not well aligned to each other – it is always advisable to raise a list of topics in the Introduction before revisiting them again in the Discussion. While bio- and phylogeography plays an important part in the Discussion, such topics are not covered al all in the Introduction. Related to this, the start of the Introduction straight goes into the taxonomy and distribution of Hynobius, whereas initially covering a more general framework of the study (e.g. on phylogeny or biogeography) would attract a wider audience to the work.

4. Some important terms needs to be treated with more care. Given the obvious, demonstrated morphological differences between the new species and existing taxa (also figure visualises these marked differences), is it really appropriate to refer to it as “cryptic” in the title? This term is generally used for species which are difficult or impossible to identify based on morphology alone. I am also a little uncomfortable with the term “species complex” for the taxa under study – which encompassed only a pair of species prior to the present work, which were also historically not always assigned to a single grouping.

5. The figures are of particularly high standard – it is very evident how much work went into preparing them. Nevertheless it is a little questionable whether all 21 display items (14 figures and 7 tables, some of them very large) are really necessary in the main body of the paper, and whether some of them could be moved to a supplement/appendix. This might bring the key messages across in a more concise way.

Experimental design

I am not an expert with all the analytical pathways applied, but the research question is well designed, and the Methods are described in sufficient detail. The data analysis follows established procedures, and appears overall sound. I only have the following two comments:

1. The sample size is on the low size for some of the inferences. While sample sizes do not appear to violate any of the conclusions drawn, low numbers should become more acknowledged throughout the paper. For example, while Hynobius boulengeri plays an import part of the work (is one of the members of the “species complex” as termed by the authors), whereas actually only three individuals were available. For the morphological analyses, a range of parametric approaches was used (t-tests, ANCOVA), and I failed to find any information on whether data based on low sample sizes were normally distributed. Also, as far as I understand it, sufficient data to discriminate between males and females were only available for a subset of populations – how did this affect the overall analyses given that a fair degree of sexual size dimorphism was found? Given the extent of the manuscript I also might have missed the information provided.

2. In the Methods chapter, morphological aspects are described before genetic aspects and analyses, whereas in the Results chapter genetics is covered before morphology. It would be better if the order would be identical in both chapters.

Validity of the findings

I have no major criticism here. The conclusions are overall well stated, but (as outlined above) a little repetitive and not fully linked to the research question as outlined in the Introduction, where phylogeographic inferences are not well covered.

The case in support for the description of a new species comes convincingly across, and all formal requirement for species descriptions appear to have been adhered to.

Additional comments

No further comments needed.

Reviewer 2 ·

Basic reporting

High quality, but too verbose, see below.

Experimental design

no comment

Validity of the findings

no comment

Additional comments

This is a thorough and valuable integrative taxonomic analysis of Hynobius kimurae-boulengeri species complex. The study has clearly defined goals, is well justified, and makes a convincing case for the species status of H. fossigenus and provides a comprehensive taxonomic description of this species. The work is obviously based on impressive material and years of expertise. I don’t have any serious criticism regarding quality of science, but I feel that a major limitation of the paper is its length. While with online-only publication the length of the paper is not an issue from a technical/publishing side, it is still a problem from a point of view of the reader. In the case of this paper a rather straightforward message is communicated in an excessively verbose manner with large amount of descriptive material in the text. That imposes a heavy burden on the reader.
Specific comments
Abstract is too long and contains unnecessary details, such as: the length of sequenced DNA fragments, details about statistical and phylogenetic methods, sample sizes for individual analyses etc. Briefly, I would suggest what you did and what you found while limiting methodological details. Time of divergence estimates values with two decimals make false impression of precision of the order that is not achievable with molecular clock analyses.
l. 68-95 this history of research and a plethora of names makes a rather tedious read and disrupt the flow of the ms. Could you please consider moving it to an appendix or supplementary materials?
l. 150 a single nuclear gene is a very weak (weakly informative) marker of gene flow, especially as only a single individual of H. boulengeri was used; please make sure that you discuss limitations of using only a single nuclear marker
l. 242-46 ratios are rather questionable taxonomical tool. I have impression that you put too much emphasis on them Why don’t you rely mainly on PCA, which takes all the variables into account simultaneously
l. 258 ml -> ul
l.314-320 here you describe the procedure to deal with heterozygous positions but later in results (l. 348, should “heterogeneous” be “heterozygous”?) you say that there were none
l. 346 what you mean by “conservatie”? not variable?
376-392 please shorten this part, perhaps this information could be added to the figure showing the distribution
l. 402-420 this part is too long, and I don’t think that interpreting these results as mito-nuclear discordance is warranted with only a single nuclear marker of low variability
l. 435-454 most of this descriptive material could be moved to tables
781 brakes -> breaks
840-918 this comparison is descriptive and very long, it relies heavily on ranges of morphological indices which is questionable from both theoretical and practical point of view

Measurements, i.e. raw data should be in supplementary tables, not in the main text

Reviewer 3 ·

Basic reporting

In an attached annotated PDF, I have provided additional minor corrections to grammar, spelling, and typos, and suggest some word changes and rephrasing. Throughout the manuscript, there are numerous minor grammatical errors (e.g., omissions of "the") that I assume are due to nonnative English language. But I strongly believe that academic publishing should be globally inclusive and as accommodating as possible to scientists whose native language is not English, so I do not suggest the authors hire a proofreader or spend an excessive amount of time correcting very minor errors that do not in any way hinder clarity or understanding of the manuscript. Therefore, I did not mark any of those extremely minor errors; I primarily made corrections/suggestions for errors or phrasing that made the authors' meaning unclear. Overall, I was quite impressed with the writing!

Minor but still important comments & suggestions:

- Would it be preferable to include the new species name Hynobius fossigenus in the title?
- Abstract: too much detail. Abstracts should be short and concise.
- L. 39: reword to omit "basal" -- this misuse of the term "basal" is very common in the phylogenetics literature, unfortunately. I suggest something like "The H. kimurae - H. boulenteri species complex is sister to all other Hynobius except H. retardus." (To compare: Table 3 is a correct use of "basal" because it is referring to ancestral nodes, not lineages or extant species.)
- L. 442-454: Is it necessary to present the same data here and also in the new species diagnosis?
- L. 472-506: I'm not familiar enough with papers that include new species descriptions to determine what information belongs in this section vs. the Discussion below, and what information (if any) should be duplicated and presented both here and in the Discussion. Some of this seems repetitive, but it's A LOT of information, so it's hard for me to follow and really determine what is repetitive or not.
- Fig. 2: it is very difficult to see gray vs. white node markers. I suggest completely eliminating the circle markers on unsupported nodes and changing the gray circles to white (i.e., black = strong support, white = moderate support). Also, specifically define the cutoffs you use for strong vs. moderate support.
- Fig. 3: Does the gray shading on the inset paleogeographic reconstructions indicate above-water land? This should be added to the legend.

Very minor comments & suggestions:

- L. 90: define the abbreviation "mtDNA" at its first use.
- L. 231-232: "presence of the whiptail-like structure (WT) and the following five characters were recorded" - confusing wording. Why is WT listed separately from the other 5 characters?
- L. 256: "mtDNA" already defined above
- L. 342-344: confusing wording
- L. 354 and forward: Seems strange to explicitly state e.g., "Results of phylogenetic analyses are shown in Fig. 2." Usually, figures and tables are referred to parenthetically within the sentence discussing results shown in that figure/table.
- L. 389: italicize species name.
- L. 394-395 and caption for Fig. 2: reword. I don't think you mean to say you calculated genetic distance between 16S and cytb.
- L. 418: reword. The data don't just "suggest a possibility" of this; they do show mito-nuclear discordance.
- L. 435-454: It's cumbersome and a bit confusing the way these populations are referred to in this section. I suggest using either only the locality name (e.g., "the Mt. Hinode population") or "western clade" and "eastern clade."
- L. 456-457: I've most often seen PCA factors referred to as "PC1" and "PC2." The terms "F1" and "F2" in biological literature usually refer to offspring generations.
- L. 576 and forward: Seems strange to always spell out the full genus name of Hynobius fossigenus instead of abbreviating to H. fossigenus as appropriate, especially outside of the species description/diagnosis. I suggest changing this unless it is standard practice in papers that include a new species description.
- L. 597: Is it normal for genetic diagnoses to be this vague and subjective? I.e., does more specific information from the Results section need to be repeated here or not? I am not a nomenclature expert...
- L. 795-796: confusing wording. Do you mean the range of H. fossigenus (which would not make sense), or the combined ranges of all species in the genus?
- L. 827-832: This is a confusing run-on sentence that should be split into 2 or more complete sentences.
- L. 840-843: confusing wording.
- L. 1013-1016: awkward run-on(?) sentence.
- L. 1029-1035: This is just restating information already presented in Results and Discussion and doesn't need to be repeated here.
- Fig. 10: "egg sacs" is hyphenated throughout the body of the manuscript but is not hyphenated in the legend for Fig. 10. I have most often seen the term as two words, not hyphenated, but this is a very minor issue.
- Table 4: I suggest changing the headings "Min-Max" to "Range" to clarify that this should not be interpreted as the maximum subtracted from the minimum, and also because the values in those columns are referred to as "range" in the figure legend. Use of "Min-Max" alongside "Mean±SD" is a bit confusing.
- Tables 5-7: Should these tables be moved to Supplementary Information?

Experimental design

Major issue:

L. 90-94: I'm confused -- Weisrock et al. 2013 does not show H. kimurae to be sister to H. boulengeri, but instead sister to H. hirosei, and H. boulengeri is quite distantly related. In contrast, Nishikawa & Matusi 2014 show H. kimurae sister to H. boulengeri, and H. hirosei is more distant. (I don't have access to Larson et al. 2003.) As far as I can tell, Weisrock et al. 2013 and Nishikawa & Matusi 2014 both included H. hirosei from Shikoku Island, H. boulengeri from Kii Peninsula, and H. kimurae from Shiga (within western clade [H. kimurae sensu stricto] in this manuscript). The only difference is that Weisrock et al. 2013 sequenced 12S-16S and ND2-COI, whereas Nishikawa & Matusi 2014 sequenced cytb. Is there another explanation for the discordance between the two phylogenies? A brief discussion of the conflicting phylogenies from Weisrock et al. 2013 and Nishikawa & Matusi 2014 should be mentioned, especially since Weisrock et al. 2013 does not show H. kimurae to be sister to H. boulengeri. Also, I think it is important to include separate BI phylogenies for cytb and 16S, in addition to the concatenated data set, since these prior publications show very different phylogenies. The figures for phylogenies of cytb alone and 16S alone could be added to Supplementary Information, and a brief statement added to the Results...and to the Discussion if it turns out the two phylogenies are not concordant. Though, even when lotic/lentic is plotted on the phylogeny from Weisrock et al. 2013, the statement on L. 941-945 that stream-breeding adaptations have evolved multiple times still holds.

Minor but important comments & suggestions:

- L. 236: Please explain why only this 1 population.
- L. 288-289: Why were these additional sequences for H. kimurae and H. boulengeri included? E.g., were they samples from unique localities?
- L. 292-294: Please explain here the purpose of this analysis with an expanded data set. From the current wording of this paragraph, it is not entirely clear that this is a separate data set and separate analysis from the data set and analysis described above in L. 289-290.

Very minor comments & suggestions:

- L. 295-301: Confusing wording. I assume you first concatenated the data, then ran PartitionFinder, then determined from PartitionFinder results what your partitions should be. The phrases/sentences here should be reordered.

Validity of the findings

Minor but still important comments & suggestions:

- L. 730-731: "clearly corresponds to a species level of differentiation" -- be careful how strongly this is worded. Determining the level of genetic differentiation sufficient for elevating to species status is essentially entirely subjective, so I don't think "clearly" is an appropriate word here.

Very minor comments & suggestions:

- L. 357-359: It is often unclear what is meant by "high resolution" in phylogenetic context. Better phrasing is to note that phylogenetic relationships among major lineages were strongly supported.
- L. 363-367: Where does this information fit in with the current study? Were these species once considered part of H. boulengeri? Meaning of "former members of the H. boulengeri species complex" is unclear.
- L. 374-375: Reword. There was either moderate support or there was not. Trees don't "suggest" levels of support; the bootstrap/bpp data either support or don't support particular clades.

Additional comments

This is an interesting study on Hynobius salamanders in Japan. The authors use genetic and morphological data to show that the eastern clade of "H. kimurae" should be elevated to a new species, and they provide and thorough diagnosis and description of this proposed new species, Hynobius fossigenus. This study used only mitochondrial genes and 1 nuclear gene, but within their focal species complex the mitochondrial and genetic data agree that H. fossigenus is distinct from H. kimurae (despite mito-nuclear discordance regarding relationship to H. boulengeri), and their morphological data provide additional support for elevation of the eastern clade of H. kimurae sensu lato to species status as H. fossigenus. My only major concern with this study is regarding expanded analysis and discussion of phylogenetic relationships of other Hynobius species, and ecological conclusions based on that phylogenetic inference. This study will be an important addition to the salamander literature once the concerns below are addressed.

Annotated reviews are not available for download in order to protect the identity of reviewers who chose to remain anonymous.

---

## Round 0.2 · Minor Revisions

Thank you for your revised ms and detailed response to reviewers comments. Although not reduced much in length, I'm happy to accept this version, subject to you performing a final check of grammar. For example, I noticed the following errors:

p2 "seminal", not "seminate"
p16,34 (and elsewhere?) "occurred", not "occured"
36 use "freshwater" (and elsewhere; no hyphen)

---

## Round 0.3 · accepted · Accept

Thank you for making the changes. Please change "occurrs" to "occurs" (line 697).

#